# Overcoming Recency Bias of Normalization Statistics in Continual Learning: Balance and Adaptation

**Yilin Lyu**[1][*]  **Liyuan Wang**[2][*]  **Xingxing Zhang**[2][†]  **Zicheng Sun**[1]
**Hang Su**[2]   **Jun Zhu**[2]   **Liping Jing**[1][†]
[1]Beijing Key Lab of Traffic Data Analysis and Mining, Beijing Jiaotong University
[2]Dept. of Comp. Sci. & Tech., Institute for AI, BNRist Center, THBI Lab,
Tsinghua-Bosch Joint Center for ML, Tsinghua University
{yilinlyu, zichengsun, lpjing}@bjtu.edu.cn
wly19@tsinghua.org.cn, xxzhang1993@gmail.com
{suhangss, dcszj}@tsinghua.edu.cn

## Abstract

Continual learning entails learning a sequence of tasks and balancing their knowledge appropriately. With limited access to old training samples, much of the current work in deep neural networks has focused on overcoming catastrophic forgetting of old tasks in gradient-based optimization. However, the normalization layers provide an exception, as they are updated interdependently by the gradient and statistics of currently observed training samples, which require specialized strategies to mitigate recency bias. In this work, we focus on the most popular Batch Normalization (BN) and provide an in-depth theoretical analysis of its sub-optimality in continual learning. Our analysis demonstrates the dilemma between balance and adaptation of BN statistics for incremental tasks, which potentially affects training stability and generalization. Targeting on these particular challenges, we propose Adaptive Balance of BN (AdaB$^2$N), which incorporates appropriately a Bayesian-based strategy to adapt task-wise contributions and a modified momentum to balance BN statistics, corresponding to the training and testing stages. By implementing BN in a continual learning fashion, our approach achieves significant performance gains across a wide range of benchmarks, particularly for the challenging yet realistic online scenarios (e.g., up to 7.68%, 6.86% and 4.26% on Split CIFAR-10, Split CIFAR-100 and Split Mini-ImageNet, respectively). Our code is available at https://github.com/lvyilin/AdaB2N.

## 1 Introduction

Continual learning aims to acquire, accumulate and exploit knowledge from sequentially arrived tasks. As the access to old training samples is typically limited, the acquired knowledge tends to be increasingly relevant to the more recent tasks. Such recency bias, also known as catastrophic forgetting [17], is generally attributed to the nature of gradient-based optimization in deep neural networks (DNNs) [9, 30], where the learning of network parameters depends on a tug-of-war over currently observed training samples. Numerous efforts have been devoted to addressing catastrophic forgetting of these learnable parameters through rectifying their gradients. However, as an important component of DNNs, the normalization layers employ particular updating rules that are only partially gradient-based, and their recency bias in continual learning remains to be addressed.

---

[*]Equal contribution.
[†]Corresponding authors.

37th Conference on Neural Information Processing Systems (NeurIPS 2023).

Table 1: Comparison of normalization methods for continual learning.

| Method | Statistics | Training Adaptability | Online | Offline |
|---|---|:---:|:---:|:---:|
| BN [11] | Unbalanced | ✗ | ✗ | ✗ |
| CN [19] | Unbalanced | ✓ | ✓ | ✗ |
| BNT [34] | Balanced | ✗ | ✗ | ✓ |
| TBBN [8] | Balanced | ✓ | ✗ | ✓ |
| AdaB$^2$N (Ours) | Adaptively balanced | ✓ | ✓ | ✓ |

The normalization layers typically include *affine transformation parameters* and *statistics parameters* to normalize the internal representations, in order to accelerate convergence and stabilize training [11, 33, 2, 24]. Since these two kinds of parameters are updated respectively from the gradient and statistics (i.e., the empirical mean and variance) of currently observed training samples, the normalization layers require specialized strategies to overcome recency bias in continual learning. In particular, the most popular Batch Normalization (BN) [11] performs an exponential moving average (EMA) of statistics parameters for testing, which preserves historical information to some extent, and thus outperforms many other alternatives in continual learning [19, 8]. As the BN statistics of old tasks still decay exponentially, some recent work [19, 8, 34] enforced a balance in the contribution of individual tasks to the BN statistics, but with limited effectiveness and generality (see Table 1 for a conceptual comparison of their targets).

So, how to update normalization statistics for better continual learning: *balance* or *adaptation*? In this work, we present an in-depth theoretical analysis of the most popular BN, demonstrating its potential challenges posed by continual learning. First, a regular EMA with constant momentum cannot guarantee the balance of BN statistics for testing, which yields more recency bias as more training batches of each task are observed. Although a modified EMA with normalized momentum of batch number can enforce balanced statistics, they tend to be outdated for learning each task and thus limit model adaptability. Besides, the sharp changes in task distributions can interfere with the benefit of BN in stabilization of training, which further affects continual learning.

Based on these theoretical insights, we propose Adaptive Balance of BN (AdaB$^2$N) to improve continual learning. Specifically, we leverage a Bayesian-based strategy to decide an appropriate way of accommodating task-wise contributions in normalized representations to stabilize training. We further incorporate an adaptive parameter to reconcile the constant and normalized momentum of EMA, in order to strike a balance of BN statistics for testing while training each task with adequate adaptability. Extensive continual learning experiments demonstrate the superiority of our approach, which provides substantial improvements especially for the challenging yet realistic online scenarios.

## 2  Related Work

**Continual Learning:** The major focus of this area is to address recency bias aka catastrophic forgetting in DNNs [17, 30, 28, 29]. Representative strategies are generally designed for the learnable parameters with gradient-based optimization [30], such as weight regularization [13, 1, 27], experience replay [32, 6], gradient projection [15, 20], etc. In particular, experience replay of a few old training samples has proven to be usually the most effective strategy and is applicable to large-scale applications [30, 32, 31]. However, its reliance on the old training samples may further deteriorate the imbalance and overfitting of normalization statistics. Corresponding to the real-world dynamics, continual learning gives rise to many meaningful settings and particular challenges [30]. For example, a desirable continual learning model should perform well in both task-incremental and class-incremental settings, which depend on whether the task identity is provided for testing. In contrast to offline scenarios of learning each task with multiple epochs, the model should also accommodate online scenarios of learning all tasks in a one-pass data stream, which is realistic but extremely more challenging.

**Normalization Layer:** The normalization layers have become a critical component for many representative DNN architectures, where Batch Normalization (BN) [11] is the most popular choice along with many other competitive alternatives designed for single-task learning [33, 24, 2]. However, the normalization layers require specialized strategies to perform appropriate parameter updates in continual learning. As the EMA of statistics parameters retains a certain amount of past information, BN is naturally the dominant choice for continual learning and indeed a strong baseline for overcoming

recency bias in normalization layers [19, 8]. To further improve BN, Continual Normalization (CN) [19] removed differences in task-wise distributions and then performed BN to acquire task-balanced statistics, providing benefits for online continual learning. BNT [34] and TBBN [8] constructed task-balanced batches to update BN, but only effective for offline continual learning. However, none of the current efforts can improve both online and offline scenarios [8], with the effectiveness of CN varying further with batch size in different online benchmarks. This motivates our work to provide a more in-depth investigation of BN with respect to the particular challenges of continual learning.

## 3 Analysis of Normalization Statistics in Continual Learning

In this section, we introduce the formulation of continual learning and representative normalization strategies, and then provide an in-depth theoretical analysis to diagnose the most popular BN. Additionally, a notation table is included in Appendix A, and proofs of theorems can be found in Appendix B.

### 3.1 Notation and Preliminaries

Let's consider a general setting for continual learning, where a neural network with parameters $\theta$ needs to learn a sequence of tasks from their training sets $\mathcal{D}_1, \cdots, \mathcal{D}_T$ in order to perform well on their test sets. The training set of task $t$ is defined as $\mathcal{D}_t = \{(x_{t,n}, y_{t,n})\}_{n=1}^{|\mathcal{D}_t|}$, where $(x_{t,n}, y_{t,n})$ is a data-label pair of task $t \in [1, T]$ and $|\cdot|$ denotes its amount. Since the old training sets $\mathcal{D}_{1:t-1} = \bigcup_{i=1}^{t-1} \mathcal{D}_i$ are unavailable when learning task $t$, $\theta$ tends to catastrophically forget the previously learned knowledge. An effective strategy is to save a few old training samples in a small memory buffer, denoted as $\mathcal{M} = \{\mathcal{M}_k\}_{k=1}^{t-1}$, in order to recover the old data distributions when learning the current task $t$. We denote the $m$-th training batch as $B_m$, which includes $N_t$ training samples from $\mathcal{D}_t$ and $N_k$ training samples from $\mathcal{M}_k$ for $k = 1, \cdots, t-1$. Here $N = \sum_{k=1}^{t} N_k = |B|$ denotes the batch size. The batch index $m \in (m_{t-1}, m_t]$, where $m_t = \lfloor |\mathcal{D}_{1:t}|/N \rfloor E$ is the index of the last batch of task $t$, $m_0 = 0$, and $E$ is the number of training epochs. We denote $r_m = N_t/N$ as the proportion of the current task $t$ in $B_m$, where $r_m < 1$ indicates the use of replay while $r_m \equiv 1$ indicates that no replay is used. Without loss of generality, we assume that the old training samples in each batch are task-balanced, i.e., $N_1 = N_2 = \cdots = N_{t-1}$. The test set follows the same distribution as the training set, with the task identity provided in task-incremental learning (Task-IL) while not provided in class-incremental learning (Class-IL) [25].

Inside the network, the internal representation of the $\ell$-th layer of the $m$-th batch is denoted as $a_m^{(\ell)} \in \mathbb{R}^{N \times C^{(\ell)} \times D^{(\ell)}}$, where $C^{(\ell)}$ and $D^{(\ell)}$ represent the number of channels and feature dimensions, respectively. For the sake of notation clarity, we omit the layer and batch index hereafter when no confusion arises. Then, the operation of a normalization layer can be generally defined as:

$$\hat{a} = \gamma a' + \beta, \qquad a' = \frac{a - \mu(a)}{\sqrt{\sigma^2(a, \mu(a)) + \varepsilon}}, \tag{1}$$

where the statistics parameters (i.e., the empirical mean and variance) are calculated from the corresponding functions $\mu(\cdot)$ and $\sigma^2(\cdot, \cdot)$, respectively, $\varepsilon$ is a small positive constant to avoid the denominator being zero, $\gamma$ and $\beta$ are affine transformation parameters that are learned by gradient-based optimization. Representative normalization strategies differ mainly in the ways of computing statistics parameters of internal representations, i.e., the exact forms of $\mu$ and $\sigma^2$, as detailed below.

**Batch Normalization** (BN) [11] is the most popular choice for a wide range of DNN architectures. During the training phase, BN calculates the statistics parameters of internal representations for each training batch through $\mu : \mathbb{R}^{N \times C \times D} \to \mathbb{R}^C$ and $\sigma^2 : \mathbb{R}^{N \times C \times D} \times \mathbb{R}^C \to \mathbb{R}^C$, where

$$\mu(a) = \frac{1}{ND} \sum_{n=1}^{N} \sum_{d=1}^{D} a_{[n,:,d]}, \quad \sigma^2(a, \mu(a)) = \frac{1}{ND} \sum_{n=1}^{N} \sum_{d=1}^{D} (a_{[n,:,d]} - \mu(a))^2. \tag{2}$$

During the testing phase, to make the normalization independent of other samples in the same batch, BN replaces the batch statistics of Eq. (1) with the population statistics estimated by exponential moving average (EMA). For ease of notation, here we define a convenient function $S : \mathbb{R}^{N \times C \times D} \to \mathbb{R}^{2 \times C}$ to obtain the two kinds of statistics, where

$$S(\cdot) = [\mu(\cdot), \sigma^2(\cdot, \mu(\cdot))]^\top. \tag{3}$$

Then we denote the statistics of internal representation $a_m^{(\ell)}$ as $S_m := S(a_m)$, with the layer index omitted. Now the population statistics of BN can be written as

$$\hat{\mathbb{E}}[S_m] := \hat{\mathbb{E}}[S|a_1, \cdots, a_m] = (1-\eta)\hat{\mathbb{E}}[S|a_1, \cdots, a_{m-1}] + \eta S_m, \qquad (4)$$

where $\eta \in (0,1)$ is the momentum of EMA and usually set to 0.1 in practice.

There are many alternative improvements of BN designed for *single-task learning* without the use of EMA. **Group Normalization** (GN) [33] can be seen as a general version, which divides the channels into $G$ groups and then normalizes the internal representations of each group. Specifically, let $\xi : \mathbb{R}^{N \times C \times D} \to \mathbb{R}^{N \times G \times K \times D}$ be a reshaping function that groups the channels, where $G$ is the group number and $K = \lfloor \frac{C}{G} \rfloor$. Then the mean and variance of GN are obtained through $\mu : \mathbb{R}^{N \times C \times D} \to \mathbb{R}^{N \times G}$ and $\sigma^2 : \mathbb{R}^{N \times C \times D} \times \mathbb{R}^{N \times G} \to \mathbb{R}^{N \times G}$, where

$$\mu(a) = \frac{1}{KD} \sum_{k=1}^{K} \sum_{d=1}^{D} \xi(a)_{[:,:,k,d]}, \quad \sigma^2(a, \mu(a)) = \frac{1}{KD} \sum_{k=1}^{K} \sum_{d=1}^{D} (\xi(a)_{[:,:,k,d]} - \mu(a))^2. \qquad (5)$$

In particular, GN is equivalent to **Layer Normalization** (LN) [2] by putting all channels into one group ($G = 1$), and **Instance Normalization** (IN) [24] by separating each channel as a group ($G = C$). In contrast to these alternatives, **Continual Normalization** (CN) [19] is designed particularly for *(online) continual learning*, which performs GN with $\gamma = 1$ and $\beta = 0$ to remove task-wise differences and then performs BN to acquire task-balanced statistics.

## 3.2 Conflict of Balance and Adaptation

Although BN has become a fundamental component for many DNN architectures, its particular recency bias in continual learning remains under addressed. To quantitatively analyze the task-wise contributions in BN statistics, we define a statistical weight (of a certain batch) of each task. Then we diagnose the recency bias of BN statistics with a specified dilemma between balance and adaptation.

**Definition 1** (Statistical weight of a task). *The statistical weight of task $t$ w.r.t. the population statistics $\hat{\mathbb{E}}[S_m]$ is the weight $w_t \in \mathbb{R}^+$ such that $w_t = \sum_i^m w_i^t / Z$. $w_i^t \in \mathbb{R}^+$ is the statistical weight of task $t$ of the $i$-th batch such that $\sum_{t=1}^{T} \sum_{i=1}^{m} w_i^t S_i^t = \hat{\mathbb{E}}[S_m]$, where $S_i^t := S(a_{i, \mathbf{1}[(\cdot, y_n) \in \mathcal{D}_t]})$.[3] $Z$ is the normalizing constant $Z = \sum_t^T \sum_i^m w_i^t$.*

We now give rigorously the statistical weight for EMA in order to demonstrate the recency bias:

**Theorem 1** (Statistical weight for EMA). *Let $T, Z$ be defined as above, and $\{m_t\}_{t=1}^{T}$ be an increasing sequence (i.e., $m_t < m_{t+1}$), where $m_t \in \mathbb{Z}^+$, $\eta : [1, m_T] \to (0,1)$, and $r : [1, m_T] \to (0,1]$. The statistical weight of task $t \in [1, T]$ w.r.t. the population statistics $\hat{\mathbb{E}}[S_{m_T}]$ estimated by Eq. (4) is*

$$w_t = \left[ \sum_{i=m_{t-1}+1}^{m_t} \eta_i r_i \prod_{j=i+1}^{m_T} (1 - \eta_j) + \sum_{i=m_t+1}^{m_T} \frac{\eta_i(1-r_i)}{T-1} \prod_{j=i+1}^{m_T} (1 - \eta_j) \right] / Z, \qquad (6)$$

*where we define $\eta_i := \eta(i)$ and $r_i := r(i)$ for brevity.*

Eq. (6) specifies the impact of each task on the EMA population statistics, which exposes **a dilemma between balance and adaptation**, as we demonstrate below.

**Corollary 2** (Adaptation of BN statistics). *Let $\{m_t\}_{t=1}^{T}$ be defined as above. If $r(\cdot) \equiv 1$ and $\eta(\cdot) \equiv \eta = 1 - \bar{\eta}$, the statistical weight of task $t \in [1, T]$ w.r.t. the population statistics $\hat{\mathbb{E}}[S_{m_T}]$ is $w_t = \frac{\bar{\eta}^{m_T - t} - \bar{\eta}^{m_T - t + 1}}{1 - \bar{\eta}^{m_T}} / Z$. When $m_T - m_{T-1} = m_{T-1} - m_{T-2} = \cdots = m_1$, we have $w_t \approx \bar{\eta}^{m_1(T-t)} / Z$, with approximation error $|\epsilon| \leq \mathcal{O}(\frac{\bar{\eta}^{m_1}}{1 - \bar{\eta}^{m_1 T}})$.*

Corollary 2 shows that without the use of replay, a model dealing with evolving task distributions has a strong preference for the distribution of the current task, as the statistical weights of past tasks decrease exponentially. Replay of old training samples can balance statistical weights of all seen tasks to some extent. However, it cannot eliminate the recency bias except in an *unrealistic* case:

---

[3] For simplicity, we assume zero covariance among the internal representations of different tasks.

**Corollary 3** (Balance of BN statistics). *Let $\{m_t\}_{t=1}^{T}$ be defined as above. If $r(\cdot) \equiv r$, $\eta(\cdot) \equiv \eta = 1 - \bar{\eta}$, and $m_T - m_{T-1} = m_{T-1} - m_{T-2} = \cdots = m_1$, then the statistical weight of each task $t \in [1, T]$ w.r.t. the population statistics $\hat{\mathbb{E}}[S_{m_T}]$ achieves (approximately) equal if and only if $r \approx 1/T$, with approximation error $|\epsilon| \leq \mathcal{O}(\bar{\eta}^{m_1})$.*

Corollary 3 shows that it is possible to balance the statistical weights by constructing a task-balanced batch with the memory buffer. However, this will severely slow down the convergence. Besides, as the memory buffer is usually limited in size, the model can easily overfit to the preserved old training samples [4], which is detrimental to generalization. Corollary 3 also demonstrates the importance of keeping flexibility of $\eta$. For instance, if we substitute $\eta(i) = 1/(1 + i)$ into Eq. (4), this leads to a normalized momentum of batch number. In this case the population statistics can be considered as the cumulative moving average (CMA) [19] of the batch statistics, which can balance the statistical weights as below:

**Corollary 4** (CMA). *Let $\{m_t\}_{t=1}^{T}$ be defined as above. If $r(\cdot) \equiv r$, then the statistical weight of each task $t \in [1, T]$ w.r.t. the population statistics $\hat{\mathbb{E}}[S_{m_T}]$ achieves equal if $\eta(i) = 1/(1 + i)$.*

However, the balanced statistical weights do not necessarily lead to better performance, as the statistics of past tasks may be outdated relative to the current model. This is empirically validated in [19], where CMA does not outperform EMA in continual learning. Therefore, an improved strategy of updating population statistics for testing is urgently needed to reconcile balance and adaptation.

In addition to the update of population statistics during the *testing* phase, the current strategy of calculating batch statistics during the *training* phase is also sub-optimal for continual learning, as detailed in the next subsection.

### 3.3  Disturbed Training Stability

BN has been shown to stabilize training and improve generalization under traditional i.i.d. data setting [3, 21, 16]. However, it is in fact not clear whether BN can achieve the same benefit in continual learning, especially when the model has just switched to a new task with drastic changes in loss and parameters. We perform a preliminary analysis by the following theorem:

**Theorem 5** (Theorem 4.1 of [21], restated). *Let $\mu, \sigma^2, \gamma, \ell, a, a'$ be defined as above. Let $\mathcal{L}$ be the loss function of a BN model (i.e., using the normalized representations via Eq. (1)) and $\mathcal{L}'$ be the identical loss function of an identical non-BN model (i.e., $a' \equiv a$). The gradient w.r.t. the internal representation $a$ of the $\ell$-th layer follows that*

$$\|\nabla_{a_{[:,j]}}\mathcal{L}\|^2 \leq \frac{\gamma^2}{\sigma_{[:,j]}^2 C} \left[ C\|\nabla_{a_{[:,j]}}\mathcal{L}'\|^2 - (\mathbf{1}^\top \nabla_{a_{[:,j]}}\mathcal{L}')^2 - (\nabla_{a_{[:,j]}}^\top \mathcal{L}' \cdot a'_{[:,j]})^2 \right]. \tag{7}$$

Theorem 5 demonstrates that BN can contribute to reducing the Lipschitz constant of the loss and thus stabilizing the training of a single task. However, as the model underfits to the distribution of each continual learning task, the upper bound of the gradient magnitude will be loosened by the increase of $\{\nabla_a^\top \mathcal{L}' \cdot a'\}$ within the third term, i.e., the similarity of the normalized representation and its corresponding gradient, which potentially affects the benefit of BN. Here we present empirical results to validate the above claim. As shown in Fig. 1, a dramatic change in loss $\mathcal{L}$ occurs whenever the task changes. We observe an increase of cosine similarity between the gradient and the normalized representation at this time, accompanied by a dramatic change in the gradient magnitude. This clearly demonstrates the importance of properly adapting BN statistics to each continual learning task, so as to better leverage the benefit of BN in stabilization of training.

## 4  Method

Following the theoretical analysis, we propose Adaptive Balance of BN (AdaB$^2$N) from two aspects: (1) **training aspect**: we apply a Bayesian-based strategy to adjust inter-task weights and construct the normalized representation, allowing for learning an appropriate way of weighting task distributions to stabilize training; and (2) **testing aspect**: instead of using a fixed momentum $\eta$ in EMA, we turn to a generalized version that balances the statistical weights of past tasks while reducing the impacts of past batches. We now present both ideas as follows with a pseudo-code in Algorithm 1.

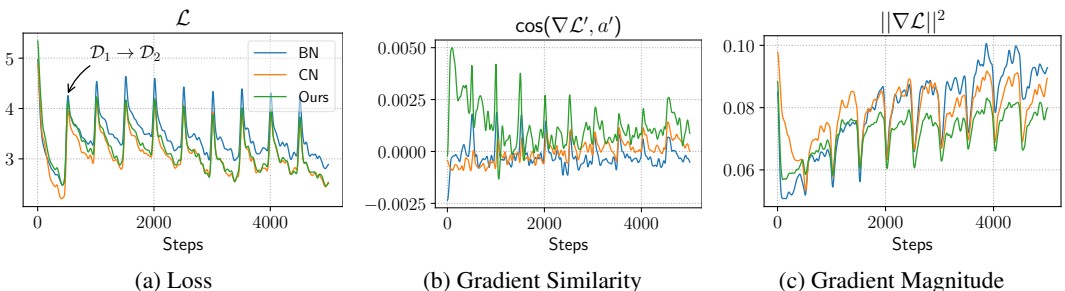

(a) Loss          (b) Gradient Similarity          (c) Gradient Magnitude

Figure 1: Dynamics of the loss of training batches, gradient similarity (i.e., cosine similarity between gradients and normalized representations) and gradient magnitude.

---

**Algorithm 1** Training Algorithm of Adaptive Balance of BN (AdaB$^2$N)

---

**Input**: training dataset $\mathcal{D}_{1:T}$, memory buffer $\mathcal{M}$, batch size $N$, neural network $f_{\vartheta,\omega} = f_{\vartheta^{(L+1)}} \circ f_{\omega^{(L)}} \circ f_{\vartheta^{(L)}} \cdots \circ f_{\omega^{(\ell)}} \circ f_{\vartheta^{(\ell)}} \cdots \circ f_{\omega^{(1)}} \circ f_{\vartheta^{(1)}}$, where $f_{\vartheta^{(\ell)}}$ and $f_{\omega^{(\ell)}}$ are the $\ell$-th network layer and normalization layer, respectively, $\omega = \{\gamma^{(\ell)}, \beta^{(\ell)}, \psi^{(\ell)}\}_{\ell=1}^{L}, \vartheta = \{\vartheta^{(\ell)}\}_{\ell=1}^{L+1} = \theta \setminus \omega$, initial parameters $\vartheta_0, \omega_0$, the objective function $\mathcal{L}$ in Eq. (15), and hyperparameters $\tilde{\eta}, \kappa, \lambda$.
**Output**: parameters $\vartheta, \omega$

---

1: Initialize: $\vartheta \leftarrow \vartheta_0, \omega \leftarrow \omega_0, \hat{\mathbb{E}}[S] \leftarrow \mathbf{0}$
2: **for** $t \leftarrow 1$ to $T$ **do**                                                          ▷ For each task
3:     **for** $e \leftarrow 1$ to $E$ **do**                                        ▷ For each epoch
4:         **for** $i \leftarrow 1$ to $\lfloor |\mathcal{D}_t|/N \rfloor$ **do**        ▷ For each batch
5:             $B_i = (X_i, Y_i) \leftarrow \text{Sample}(\mathcal{D}_t, \{\mathcal{M}_k\}_{k=1}^{t-1})$
6:             $\mathcal{M} \leftarrow \text{UpdateMemory}(\mathcal{M}, B_i)$
7:             $\hat{a}^{(0)} \leftarrow X_i$
8:             **for** $\ell \leftarrow 1$ to $L$ **do**                ▷ For each layer
9:                 $a^{(\ell)} \leftarrow f_{\vartheta^{(\ell)}}(\hat{a}^{(\ell-1)})$
10:                 Compute $\hat{\mathbb{E}}[S|a^{(\ell)}]$ via Eq. (13) and $\eta_i$ via Eq. (16)
11:                 Update $\hat{\mathbb{E}}[S]$ via Eq. (4)
12:                 Compute $a'^{(\ell)}$ via Eq. (8) and $\hat{a}^{(\ell)}$ via Eq. (1)
13:             $\hat{Y}_i = f_{\vartheta^{(L+1)}}(\hat{a}^{(L)})$
14:             $\vartheta \leftarrow \vartheta - \nabla_\vartheta \mathcal{L}, \omega \leftarrow \omega - \nabla_\omega \mathcal{L}$
15: **return** $(\vartheta, \omega)$

---

### 4.1 Adaptive Balance of BN Statistics for Training

Theorem 5 demonstrates that the benefit of BN for optimization and generalization is potentially interfered in continual learning. To overcome this challenge, here we propose a Bayesian-based strategy for learning an appropriate way of weighting task contributions in normalization statistics. We assume the statistics of batch $m$ for a certain layer follow a probability distribution $S \sim \mathcal{P}_\theta(S|a_m)$, where the randomness originates from the neural network parameters $\theta$, and then use the conditional expectation w.r.t. the distribution of statistics to calculate the normalized representation:

$$a'_m = \frac{a_m - \mathbb{E}[\mu|a_m]}{\sqrt{\mathbb{E}[\sigma^2|a_m] + \varepsilon}}. \tag{8}$$

Specifically, $\mathbb{E}[\mu|a_m]$ and $\mathbb{E}[\sigma^2|a_m]$ are the two elements of $\mathbb{E}[S|a_m]$ as described in Eq. (3):

$$\mathbb{E}[S|a_m] = \sum_S \sum_t \mathcal{P}_\theta(S|t, a_m)\mathcal{P}_\theta(t|a_m)S^t, \tag{9}$$

where $\mathcal{P}_\theta(S|t, a_m)$ is the distribution of statistics given the task identity, and $\mathcal{P}_\theta(t|a_m)$ is the likelihood of the task identity occurring in $a_m$. We treat $\mathcal{P}_\theta(t|a_m)$ as a categorical distribution over the task identity, i.e., $\mathcal{P}_\theta(t|a_m) = N_t/N$. Then, BN obtains its normalization statistics by the point estimate of the conditional expectation:

$$\hat{\mathbb{E}}[S|a_m] = \sum_t \mathcal{P}_\theta(t|a_m)S_m^t = S_m. \tag{10}$$

As analyzed in Section 3.3, it is critical to consider adaptability of the model parameters $\theta$ together with normalization statistics in model training, so as to enhance the gradient-representation correlation. Inspired by the theoretical works in Bayesian Inference [22, 12], we propose to capture the variability of $\theta$ through modeling the task distribution with a Bayesian principle, thus enhancing the benefit of BN for continual learning. Specifically, we assume that there is a random variable $\tau \in \Delta^{T-1}$ modeling the probability distribution of tasks, where $\Delta^{T-1}$ denotes the $T-1$ simplex, i.e., $\sum_t^T \tau_t = 1$ and $\tau_t > 0, \forall t$. Applying the Bayes' rule, we have

$$\mathbb{E}[S|a_m] = \sum_S \sum_\tau \sum_t \mathcal{P}_\theta(S|\tau, t, a_m)\mathcal{P}_\theta(\tau|a_m)\mathcal{P}_\theta(t|\tau, a_m)S^t, \tag{11}$$

where $\mathcal{P}_\theta(S|\tau, t, a_m)$ and $\mathcal{P}_\theta(t|\tau, a_m)$ are defined analogous to $\mathcal{P}_\theta(S|t, a_m)$ and $\mathcal{P}_\theta(t|a_m)$ in Eq. (9). $\mathcal{P}_\theta(\tau|a_m)$ serves as the prior captured from the model parameters $\theta$, which is generally intractable. For ease of computation, we approximate it with a Dirichlet distribution, i.e., the conjugate prior of the categorical distribution, which is parameterized by the concentration parameter $\phi \in \mathbb{R}^{T+}$:

$$\mathcal{P}_\phi(\tau|a_m) = \frac{\Gamma(\bar{\phi})}{\prod_t \Gamma(\phi_t)} \prod_{t=1}^T \tau_t^{\phi_t - 1}, \tag{12}$$

where $\Gamma(\cdot)$ is the Gamma function, and $\bar{\phi} = \sum_t \phi_t$. By taking advantage of algebraic convenience of the conjugate distribution, we can estimate the conditional expectation by a closed-form expression

$$\hat{\mathbb{E}}[S|a_m] = \sum_\tau \sum_t \mathcal{P}_\phi(\tau|a_m)\mathcal{P}_\theta(t|\tau, a_m)S_m^t = \sum_t \frac{\phi_t + N_t}{\bar{\phi} + N} S_m^t. \tag{13}$$

Intuitively, the concentration parameter $\phi$ plays the role of adding pseudo-observations to the task weighting procedure, which allows the normalization statistics to adaptively balance task-wise contributions by considering the current state of $\theta$. To optimize $\phi$, we introduce a learnable parameter $\psi \in \mathbb{R}^T$ and obtain $\phi = \exp(\psi)$. It is worth noting that the dimension of $\phi$ is incremental with the number of seen tasks, so there is no need to know the total number of tasks in advance. As $\phi$ is expected to capture the information of $\theta$, we propose a regularization term to minimize the Euclidean distance between the adaptively balanced statistics and the estimated population statistics via Eq. (4) as follows:

$$\mathcal{L}_{ada}^{(\ell)}(\theta, \psi) = \|\hat{\mathbb{E}}[S|a_m] - \hat{\mathbb{E}}[S_m]\|^2, \tag{14}$$

where $\hat{\mathbb{E}}[S_m]$ is an improved version introduced in Section 4.2. Eq. (14) provides $\phi$ with historical information across batches by aligning the population statistics, which contributes to obtaining statistics as good as joint training (detailed in Section 5). From Fig. 1 we observe that adding this regularization term can significantly improve the gradient similarity, leading to stabilized training dynamics. We optimize $\psi$ and $\theta$ (with $\gamma$ and $\beta$ inside) simultaneously, and the final objective function can be written as

$$\min_{\theta, \psi} \mathcal{L}_{CL}(\theta) + \lambda \sum_\ell \mathcal{L}_{ada}^{(\ell)}(\theta, \psi), \tag{15}$$

where $\mathcal{L}_{CL}$ is the loss function of continual learning (e.g., a cross-entropy loss over $\mathcal{D}_t$ and $\{\mathcal{M}_k\}_{k=1}^{t-1}$ for classification tasks). $\lambda$ is a hyperparameter to balance the two terms.

## 4.2 Adaptive Balance of BN Statistics for Testing

Inspired by Corollaries 2 to 4, we propose to employ an appropriate $\eta$ function to strike an adaptive balance between EMA and CMA. Concretely, we design a modified momentum $\eta : [1, m_T] \times (0, 1) \to (0, 1)$ satisfying the following recurrence relation:

$$\eta_i := \eta(i, \eta_{i-1}) = \frac{\eta_{i-1}}{\eta_{i-1} + (1 - \tilde{\eta})^\kappa}, \quad \eta_0 := \tilde{\eta}^\kappa, \tag{16}$$

where $\tilde{\eta} \in (0, 1)$ is the modified momentum of EMA and $\kappa \in [0, 1]$ is an additional hyperparameter to control the balance of normalization statistics. In particular, Eq. (16) degenerates to EMA when $\kappa \to 1$ and to CMA when $\kappa \to 0$. This allows us to find an inflection point for adaptive balance of statistical weights, so as to enjoy the benefits of both EMA and CMA.

## 5 Experiments

In this section, we first briefly introduce the experimental setups of continual learning, and then present the experimental results to validate the effectiveness of our approach.

Table 2: Performance of **online task-incremental learning** with batch size $|B| = 10$. We report the final average accuracy of all seen tasks (↑) with ± standard deviation.

| Method | Split CIFAR-10 | | Split CIFAR-100 | | Split Mini-ImageNet | |
|---|---|---|---|---|---|---|
| | $|\mathcal{M}|$=500 | $|\mathcal{M}|$=2000 | $|\mathcal{M}|$=2000 | $|\mathcal{M}|$=5000 | $|\mathcal{M}|$=2000 | $|\mathcal{M}|$=5000 |
| ER-ACE w/ BN | 86.34±2.35 | 89.17±1.21 | 61.21±1.63 | 63.83±1.94 | 63.62±3.14 | 64.60±1.73 |
| ER-ACE w/ LN | 78.01±6.78 | 81.59±4.39 | 42.29±0.29 | 44.32±0.35 | 45.91±1.63 | 45.82±1.91 |
| ER-ACE w/ IN | 84.57±1.42 | 86.03±1.99 | 49.46±2.25 | 50.15±1.71 | 41.07±1.66 | 40.93±4.38 |
| ER-ACE w/ GN | 81.46±3.74 | 82.85±2.14 | 44.84±1.41 | 42.50±2.06 | 43.45±3.29 | 45.43±0.62 |
| ER-ACE w/ CN | 88.32±1.43 | 90.01±0.95 | 61.00±1.58 | 63.42±0.65 | 64.23±1.22 | 65.14±1.48 |
| ER-ACE w/ Ours | **88.74±1.77** | **90.84±2.01** | **63.88±1.58** | **67.01±2.90** | **66.78±1.91** | **68.03±0.41** |
| DER++ w/ BN | 87.61±1.67 | 90.42±1.83 | 65.53±1.17 | 66.23±0.94 | 61.70±2.40 | 61.28±1.29 |
| DER++ w/ LN | 81.35±4.25 | 82.80±5.48 | 43.24±0.86 | 44.42±2.90 | 40.05±3.67 | 37.64±4.26 |
| DER++ w/ IN | 84.81±2.99 | 87.04±2.60 | 47.39±1.46 | 49.17±2.13 | 32.88±1.16 | 35.23±6.11 |
| DER++ w/ GN | 82.05±1.24 | 83.34±3.34 | 43.54±1.47 | 45.12±1.15 | 40.65±1.76 | 38.26±3.26 |
| DER++ w/ CN | 86.92±0.89 | 89.75±0.76 | 66.20±0.38 | 67.39±1.88 | 65.09±1.76 | 66.14±1.40 |
| DER++ w/ Ours | **90.15±2.52** | **91.99±0.81** | **70.33±0.49** | **71.70±0.84** | **66.42±3.62** | **69.12±2.51** |

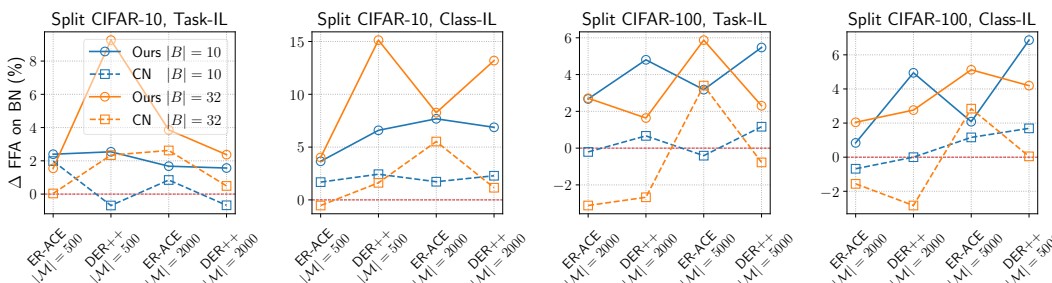

Figure 2: Performance improvements (i.e., $\Delta$ FAA) for online continual learning over different batch sizes $|B|$ and memory sizes $|\mathcal{M}|$.

**Benchmark**: We mainly focus on the setting of online continual learning [30], which is extremely challenging but realistic in application, and further consider the setting of offline continual learning. Here we evaluate three benchmarks that are widely used in both settings. Specifically, Split CIFAR-10 [14] includes 10-class images of size $32 \times 32$, randomly split into 5 disjoint tasks with 2 classes each. Split CIFAR-100 [14] includes 100-class images of size $32 \times 32$, randomly split into 20 disjoint tasks with 5 classes each. Split Mini-ImageNet [26] includes 100-class images of size $84 \times 84$, randomly split into 20 disjoint tasks with 5 classes each. We consider both Task-IL and Class-IL, depending on whether the task identity is provided at test time [25].

**Baseline**: Following the previous work [19], we employ ER-ACE [7] and DER++ [6] as the baseline approaches for continual learning, both of which rely on a small memory buffer $\mathcal{M}$ to replay old training samples. We further consider LiDER [4] in offline scenarios, a recent approach for improving experience replay through regularizing Lipschitz continuity. We then compare our approach to three categories of normalization layers: (1) BN [11], which is the most popular choice that exhibits strong continual learning performance [19, 8]; (2) BN's alternatives designed for single-task learning, such as LN [2], IN [24] and GN [33] (GN can be seen as a general version of LN and IN); and (3) CN [19], which is the state-of-the-art baseline that improves BN for online continual learning.

**Implementation**: We follow the official implementation as many previous works [19, 4, 7, 6], derived from the *mammoth* repository [5]. Specifically, we use a ResNet-18 backbone and train all baselines with an SGD optimizer of learning rate 0.03 for Split CIFAR-10/-100 and 0.1 for Split Mini-ImageNet. Notably, we find the official implementation of CN [19] employs different batch sizes (either 10 or 32) for different online continual learning benchmarks. We argue that using a smaller batch size is more "online" and more challenging, as fewer training samples of the current task can be accessed at a time. Therefore, we reproduce all results of online continual learning with one epoch and a small batch size of 10, and further evaluate the effect of using a larger batch size of 32. Besides, we use 50 epochs with a batch size of 32 for offline continual learning. We adopt reservoir sampling to save

Table 3: Performance of **online class-incremental learning** with batch size $|B| = 10$. We report the final average accuracy of all seen classes ($\uparrow$) with $\pm$ standard deviation.

| Method | Split CIFAR-10 | | Split CIFAR-100 | | Split Mini-ImageNet | |
|---|---|---|---|---|---|---|
| | $|\mathcal{M}|$=500 | $|\mathcal{M}|$=2000 | $|\mathcal{M}|$=2000 | $|\mathcal{M}|$=5000 | $|\mathcal{M}|$=2000 | $|\mathcal{M}|$=5000 |
| ER-ACE w/ BN | 48.86±2.87 | 52.25±2.55 | 24.55±1.37 | 26.07±2.78 | 14.41±1.31 | 14.53±0.58 |
| ER-ACE w/ LN | 32.29±0.37 | 36.64±1.92 | 11.50±0.10 | 11.96±1.13 | 6.28±1.08 | 6.21±1.07 |
| ER-ACE w/ IN | 44.16±1.98 | 45.86±3.25 | 15.20±0.87 | 16.15±0.80 | 5.00±0.82 | 5.13±1.48 |
| ER-ACE w/ GN | 38.97±2.04 | 39.60±3.99 | 12.25±0.83 | 11.86±0.74 | 5.33±0.83 | 6.29±0.60 |
| ER-ACE w/ CN | 50.54±4.94 | 53.97±1.20 | 23.87±1.14 | 27.23±0.37 | 15.76±0.70 | 15.39±1.45 |
| ER-ACE w/ Ours | **52.51±3.04** | **59.93±4.49** | **25.39±1.40** | **28.15±3.24** | **16.41±0.96** | **17.08±1.06** |
| DER++ w/ BN | 52.85±4.34 | 57.96±2.57 | 18.84±1.25 | 17.26±1.91 | 7.04±1.47 | 6.59±0.67 |
| DER++ w/ LN | 34.08±2.59 | 35.37±6.18 | 6.42±0.95 | 6.25±0.60 | 2.94±1.08 | 2.21±0.43 |
| DER++ w/ IN | 45.21±1.88 | 50.86±2.61 | 8.52±0.27 | 8.58±0.46 | 1.88±0.33 | 2.33±0.83 |
| DER++ w/ GN | 39.54±1.47 | 38.48±1.13 | 7.44±0.78 | 6.77±0.52 | 2.80±0.18 | 2.72±0.54 |
| DER++ w/ CN | 55.28±1.06 | 60.25±3.39 | 18.84±0.09 | 18.95±1.05 | 10.16±1.07 | 8.94±0.94 |
| DER++ w/ Ours | **59.44±2.65** | **64.83±1.89** | **23.79±2.23** | **24.12±1.63** | **10.80±2.76** | **10.85±1.58** |

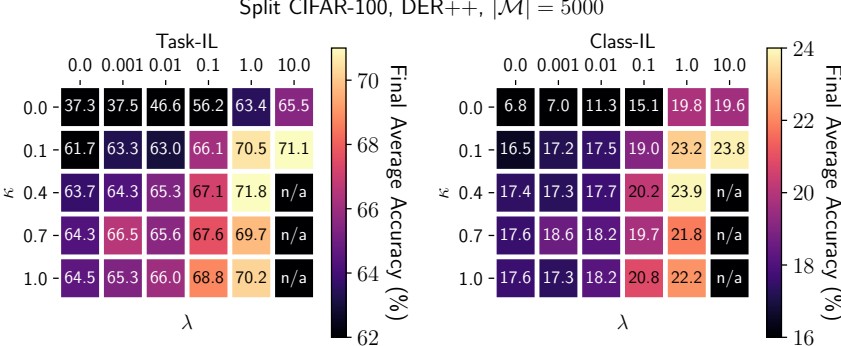

Figure 3: Hyperparameter analysis of online task-incremental learning and class-incremental learning for ablation study.

old training samples, which simulates the distribution of incoming batches, and further consider ring buffer that enforces a balanced number for each class [30]. The memory buffer is sized differently to evaluate its impact. The group number of GN and CN is set to 32 as [19]. All results are averaged over three runs with different random seeds and task splits. Please see Appendix C for more details.

**Overall Performance:** We first evaluate our approach for *online* continual learning, including the settings of online Task-IL in Table 2 and online Class-IL in Table 3. Consistent with reported results [19, 8], BN performs much better than its alternatives designed for single-task learning (i.e., GN, LN and IN), and is moderately improved by the state-of-the-art CN. In contrast, our approach can improve the performance of BN by a large margin (e.g., up to 7.68%, 6.86% and 4.26% for online Class-IL on Split CIFAR-10, Split CIFAR-100 and Split Mini-ImageNet, respectively), and outperforms CN on a wide range of *memory sizes* $|\mathcal{M}|$. When increasing the *batch size* $|B|$ from 10 to 32, which introduces more training samples at once and makes online continual learning more "offline", we observe that the benefits of CN exhibit variation in many cases, while the benefits of our approach remain consistent and become even stronger (e.g., up to 15% on Class-IL of Split CIFAR-10, see Fig. 2). Meanwhile, the *selection strategies* of the memory buffer $\mathcal{M}$ have no substantial effect, where our approach can provide a similar magnitude of improvement under either reservoir sampling or ring buffer (see Appendix D). We further present the results of *offline* Task-IL and Class-IL on the hardest Split Mini-ImageNet of the selected benchmarks (see Table 4), where CN even results in negative impacts on BN while our approach still provides strong performance gains in most cases.

**Ablation Study & Analysis:** We first analyze the introduced two *hyperparameters* in our approach, i.e., $\lambda$ of the regularization term for training and $\kappa$ of the modified momentum for testing. As shown in Fig. 3, the two hyperparameters can clearly influence the performance of continual learning, which

Table 4: Performance of **offline task-incremental learning** and **class-incremental learning** on Split Mini-ImageNet with memory size $|\mathcal{M}| = 2000$.

| Method | Task-IL | | | Class-IL | | |
|---|---|---|---|---|---|---|
| | ER-ACE | DER++ | LiDER | ER-ACE | DER++ | LiDER |
| BN | 36.36±2.33 | 35.91±1.59 | 36.96±1.92 | 10.85±0.54 | 12.69±1.23 | **10.99±0.74** |
| CN | 35.83±1.43 | 35.46±1.42 | 36.23±2.62 | 10.00±0.62 | 12.13±0.92 | 9.97±0.36 |
| Ours | **37.64±0.85** | **36.98±1.52** | **37.36±0.79** | **11.05±0.10** | **13.07±1.30** | *10.96±0.67* |

validates the effectiveness of each component in AdaB$^2$N, and obtain consistent improvements under a wide range of combinations in different settings. Besides, we visualize the *statistics parameters* of normalization layers in continual learning, in order to explicitly demonstrate how our approach can improve BN (see Fig. 4 for a typical example and Appendix D for all results). We take joint training (JT) of all tasks as the upper bound baseline for continual learning. As more training batches are introduced, it can be obviously seen that the BN statistics are increasingly deviated while the CN statistics are enforced to stabilize. They correspond to the biased strategies of *adaptation* and *balance*, respectively, but neither can fit into the curve of JT. In contrast, our approach achieves an adaptive balance of normalization statistics in continual learning, matching closely the upper bound baseline.

## 6  Discussion and Conclusion

In this work, we focus on the most popular BN in continual learning and theoretically analyze its particular challenges for overcoming recency bias. Our analysis demonstrates the dilemma between balance of all tasks and adaptation to the current task, reflected in both training and testing. Following such theoretical insights, we propose Adaptive Balance of BN (AdaB$^2$N) with a Bayesian-based strategy and a modified momentum to implement BN in a continual learning fashion, which achieves strong performance gains across various scenarios. Interestingly, the robust biological neural networks address a similar dilemma with a combination of normalization mechanisms for neurons, synapses, layers and networks, some of which are thought to be analogous to the normalization layers in DNNs [23]. More in-depth analysis and modeling of these mechanisms would be a promising direction. We also expect subsequent work to further improve continual learning of other DNN components and overcome potential incompatibilities in optimization, as suggested by our work on normalization layers.

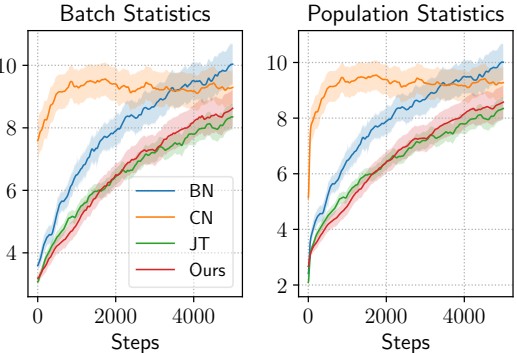

Figure 4: The norm of BN statistics of a representative layer with 0.05×variance in shaded area.

Now we discuss some potential limitations and societal impacts. First, our work applies only to the DNN architectures that employ normalization layers, especially the most popular BN. Second, we need to compute a categorical distribution $\mathcal{P}_\theta(t|a_m)$ over the task identity, which is difficult to adapt directly to the more general setting of task-agnostic continual learning. Third, the prior $\mathcal{P}_\theta(\tau|a_m)$ is approximated as a Dirichlet distribution for ease of computation, which may not be applicable in some practical scenarios. We argue that the broader impact is not obvious at the current stage, as this work is essentially a fundamental research in machine learning.

## Acknowledgements

This work was partly supported by the National Natural Science Foundation of China under Grant 62176020, 62106123; the National Key Research and Development Program (2020AAA0106800); the Joint Foundation of the Ministry of Education (8091B042235); the Beijing Natural Science Foundation under Grant L211016; the Fundamental Research Funds for the Central Universities (2019JBZ110); and Chinese Academy of Sciences (OEIP-O-202004). L.W. was also supported by Shuimu Tsinghua Scholar.

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

# A   Notation

Table 5 summarizes the notations used in this paper.

Table 5: Notations used in this paper.

| Symbol | Meaning |
|---|---|
| $T$ | Number of tasks |
| $(x_{t,i}, y_{t,i})$ | Input samples and its label of task $t$ |
| $\mathcal{D}_t$ | Training set of task $t$ |
| $\mathcal{D}_{1:t-1}$ | Old training sets before task $t$ |
| $\mathcal{M}$ | Memory buffer |
| $\mathcal{M}_t$ | Set of samples from task $t$ saved in memory buffer |
| $B, B_m$ | $m$-th batch |
| $N, |B|$ | Batch size |
| $E$ | Number of training epochs |
| $m_t$ | Index of the last batch of task $t$ |
| $r_m, r(m)$ | Proportion of task $t$ in $B_m$ |
| $\eta_m, \eta(m)$ | Momentum of exponential moving average (EMA) for $B_m$ |
| $a, a_m, a_m^{(\ell)}$ | Internal representation of the $\ell$-th layer of the $m$-th batch |
| $a', a'^{(\ell)}, a_m'^{(\ell)}$ | Normalized internal representation $a_m^{(\ell)}$ |
| $\hat{a}, \hat{a}^{(\ell)}, \hat{a}_m^{(\ell)}$ | Affined internal representation $a_m^{(\ell)}$ |
| $C, C^{(\ell)}$ | Number of channels of $a_m^{(\ell)}$ |
| $D, D^{(\ell)}$ | Feature dimensions of $a_m^{(\ell)}$ |
| $\mu(\cdot)$ | Mean function of a normalization layer |
| $\sigma^2(\cdot, \cdot)$ | Variance function of a normalization layer |
| $S(\cdot)$ | Statistic function of a normalization layer to obtain mean and variance |
| $w_t$ | Statistical weight of task $t$ |
| $w_m^t$ | Statistical weight of task $t$ within the $m$-th batch |
| $S_m$ | Batch statistics of internal representation $a_m$ |
| $S_m^t$ | Batch statistics of task $t$ within internal representation $a_m$ |
| $\hat{\mathbb{E}}[S_m], \hat{\mathbb{E}}[S|a_1, \cdots, a_m]$ | Estimated population statistics w.r.t. $a_1, a_2, \cdots, a_m$ |
| $\mathbb{E}[S|a_m]$ | Conditional expectation of batch statistics w.r.t. $a_m$ |
| $\hat{\mathbb{E}}[S|a_m]$ | Estimated conditional expectation of batch statistics w.r.t. $a_m$ |
| $\theta$ | Parameters of the entire neural network |
| $\vartheta$ | Parameters of the network layer |
| $\omega$ | Parameters of the normalization layer |
| $\gamma, \beta$ | Parameters of affine transformation |
| $\psi$ | Parameters of adaptive balance |
| $\mathcal{L}$ | Objective function |

# B   Proofs

*Proof of Theorem 1.* Let $g : [1, m_T] \times \mathbb{Z}^{T+} \to [1, T]$ be a function that maps the batch index to its corresponding task index. We abbreviate $g(i, (m_t)_{t=1}^T)$ as $g(i)$. Then we have

$$\hat{\mathbb{E}}[S_{m_T}] = (1 - \eta_{m_T})\hat{\mathbb{E}}[S_{m_T-1}] + \eta_{m_T} S_{m_T} \tag{17}$$

$$= (1 - \eta_{m_T})\hat{\mathbb{E}}[S_{m_T-1}] + \eta_{m_T} r_{m_T} S_{m_T}^T + \frac{\eta_{m_T}(1 - r_{m_T})}{T - 1} \sum_{t=1}^{T-1} S_{m_T}^t \tag{18}$$

$$= \sum_{i=1}^{m_T} \eta_i r_i \prod_{j=i+1}^{m_T} (1 - \eta_j) S_i^{g(i)} + \sum_{j=m_{g(i)}+1}^{m_T} \frac{\eta_j(1 - r_j)}{T - 1} \prod_{k=j+1}^{m_T} (1 - \eta_k) S_j^{g(j)} \tag{19}$$

where the first part of Eq. (19) can be considered as the contribution of the "current" tasks of all batches to the running statistics, and the second part can be considered as the contribution of the

"past" tasks. Now we can conclude the proof by noting that the contribution of task $t$ as the "current" tasks to the running statistics starts from $m_{t-1} + 1$ to $m_t$ and the contribution as the "past" task starts from $m_t + 1$ to $m_T$. $\qquad\square$

*Proof of Corollary 2.* The first result can be obtained by noting that $\eta$ is a constant and directly applying the sum formula of geometric series. When $m_T - m_{T-1} = m_{T-1} - m_{T-2} = \cdots = m_1$, we have that

$$w_t = \frac{\bar{\eta}^{m_{T-t}} - \bar{\eta}^{m_{T-t+1}}}{(1 - \bar{\eta}^{m_T})Z} = \frac{\bar{\eta}^{(T-t)m_1} - \bar{\eta}^{(T-t+1)m_1}}{(1 - \bar{\eta}^{Tm_1})Z} = \frac{\bar{\eta}^{(T-t)m_1}(1 - \bar{\eta}^{m_1})}{(1 - \bar{\eta}^{Tm_1})Z} \approx \frac{\bar{\eta}^{(T-t)m_1}}{Z}$$

The approximation error is upper bounded by

$$|\epsilon| = \frac{\bar{\eta}^{(T-t)m_1}}{Z}\left(\frac{1 - \bar{\eta}^{m_1}}{1 - \bar{\eta}^{m_1 T}} - 1\right) < \frac{\bar{\eta}^{m_1}}{(1 - \bar{\eta}^{m_1 T})Z} \le \mathcal{O}\left(\frac{\bar{\eta}^{m_1}}{1 - \bar{\eta}^{m_1 T}}\right)$$

$\qquad\square$

*Proof of Corollary 3.* Necessity:

Let us start by assuming that $w_t = w_{t+1}$. By applying Eq. (6) we have the following

$$\sum_{i=m_{t-1}+1}^{m_t} \eta_i r_i \prod_{j=i+1}^{m_T} (1 - \eta_j) = \sum_{i=m_t+1}^{m_{t+1}} \frac{\eta_i(r_i T - 1)}{T - 1} \prod_{j=i+1}^{m_T} (1 - \eta_j)$$

Simplifying the equation we obtain that

$$\sum_{i=m_{t-1}+1}^{m_t} r(1 - \eta)^{m_T - i} = \sum_{i=m_t+1}^{m_{t+1}} \frac{rT - 1}{T - 1}(1 - \eta)^{m_T - i}$$

$$\frac{r}{(1 - \eta)^{m_{t-1}+1}} \sum_{i=0}^{m_t - m_{t-1}} \frac{1}{(1 - \eta)^i} = \frac{rT - 1}{(T - 1)(1 - \eta)^{m_t+1}} \sum_{i=0}^{m_{t+1} - m_t} \frac{1}{(1 - \eta)^i}$$

$$\frac{r(\eta - 1)(1 - \eta)^{m_t - m_{t-1}} + r}{\eta(1 - \eta)^{m_t+1}} = \frac{(rT - 1)[(\eta - 1)(1 - \eta)^{m_{t+1} - m_t} + 1]}{\eta(T - 1)(1 - \eta)^{m_{t+1}+1}}$$

$$r(1 - \bar{\eta}^{m_t - m_{t-1}+1}) = \frac{(rT - 1)(1 - \bar{\eta}^{m_{t+1} - m_t+1})}{(T - 1)\bar{\eta}^{m_{t+1} - m_t}}$$

$$r - \frac{r(T - T\bar{\eta}^{m_{t+1} - m_t+1})}{(T - 1)\bar{\eta}^{m_{t+1} - m_t}(1 - \bar{\eta}^{m_t - m_{t-1}+1})} = \frac{\bar{\eta}^{m_{t+1} - m_t+1} - 1}{(T - 1)\bar{\eta}^{m_{t+1} - m_t}(1 - \bar{\eta}^{m_t - m_{t-1}+1})}$$

$$r = \frac{\bar{\eta}^{m_{t+1} - m_t+1} - 1}{\bar{\eta}^{m_{t+1} - m_t}(1 - T)(\bar{\eta}^{m_t - m_{t-1}+1} - 1) + T(\bar{\eta}^{m_{t+1} - m_t+1} - 1)}$$

$$= \frac{\bar{\eta}^{m_1+1} - 1}{\bar{\eta}^{m_1}(1 - T)(\bar{\eta}^{m_1+1} - 1) + T(\bar{\eta}^{m_1+1} - 1)}$$

$$= \frac{1}{T - \bar{\eta}^{m_1}(T - 1)}$$

$$\approx \frac{1}{T}$$

The approximation error is upper bounded by

$$|\epsilon| = \frac{\bar{\eta}^{m_1}(T - 1)}{T(T - \bar{\eta}^{m_1}(T - 1))} < \frac{\bar{\eta}^{m_1}}{T - \bar{\eta}^{m_1}(T - 1)} \le \mathcal{O}(\bar{\eta}^{m_1})$$

Sufficiency:

$$w_{t+1} - w_t = \sum_{i=m_t+1}^{m_{t+1}} \frac{\eta_i(r_i T - 1)}{T - 1} \prod_{j=i+1}^{m_T} (1 - \eta_j) - \sum_{i=m_{t-1}+1}^{m_t} \eta_i r_i \prod_{j=i+1}^{m_T} (1 - \eta_j) \qquad (20)$$

Substituting $r = 1/T$ into Eq. (20) yields

$$w_{t+1} - w_t = -\sum_{i=m_{t-1}+1}^{m_t} \frac{\eta_i}{T} \prod_{j=i+1}^{m_T} (1 - \eta_j)$$

$$= -\frac{\eta}{T} \sum_{i=m_{t-1}+1}^{m_t} (1 - \eta)^{m_T - i}$$

$$= -\frac{\bar{\eta}^{m_T - m_t - 1}(1 - \bar{\eta}^{m_t - m_{t-1} + 1})}{T}$$

$$\approx 0$$

The approximation error is upper bounded by

$$|\epsilon| = \frac{\bar{\eta}^{m_T - m_t - 1}(1 - \bar{\eta}^{m_t - m_{t-1} + 1})}{T} < \frac{\bar{\eta}^{m_T - m_t - 1}}{T} \leq \mathcal{O}(\bar{\eta}^{m_1})$$

$\square$

*Proof of Corollary 4.* Substituting $\eta(i) = 1/(1 + i)$ into Eq. (6)

$$w_t = \left[ \sum_{i=m_{t-1}+1}^{m_t} \eta_i r_i \prod_{j=i+1}^{m_T} (1 - \eta_j) + \sum_{i=m_t+1}^{m_T} \frac{\eta_i(1 - r_i)}{T - 1} \prod_{j=i+1}^{m_T} (1 - \eta_j) \right] / Z$$

$$= \left[ \sum_{i=m_{t-1}+1}^{m_t} \frac{r_i}{1 + m_T} + \sum_{i=m_t+1}^{m_T} \frac{1}{T - 1} \frac{1 - r_i}{1 + m_T} \right] / Z$$

$$= \left[ \frac{r + \frac{1-r}{T-1}}{1 + m_T} \right] / Z$$

We can now conclude by noting that $w_t$ is independent of the choice of $t$. $\square$

## C   Implementation Details

The hyperparameter $\tilde{\eta}$ is fixed to $0.1$, $\kappa$ is selected from $\{0.1, 0.4, 0.7, 1.0\}$, and $\lambda$ is selected from $\{0.01, 0.1, 1.0, 10.0\}$ for Split CIFAR-10 and Split CIFAR-100 and $\{0.00001, 0.0001, 0.001, 0.01\}$ for Split Mini-ImageNet. Code is available in the supplementary material and will be released upon acceptance. All experiments are performed on eight NVIDIA RTX A4000 GPUs. The amount of compute is easily affordable, which can be inferred from the running times given in Table 6.

## D   Extended Results

### D.1   Time Complexity Analysis

We provide in Table 6 the floating point operations per step (FLOPs/step) of different normalization methods and the total running times (in seconds) under different implementations (the exact calculation of FLOPs can be found in the attached code). It can be seen that compared to BN, our proposed method only slightly increases the computation, while CN almost doubles the computation because it combines both GN (without affine transformation) and BN. Note that the FLOPs only give the theoretical computation; the actual running time depends on the implementation. We measure the actual running time of two normalization implementations: 1) using the plain `torch` primitive and 2) using the `torch.nn` with cuDNN as the backend. Due to engineering difficulties, we do not currently implement a cuDNN-optimized version of our method. However, it can be seen from Table 6 that on the plain implementation our method only slightly increases the running time, which is consistent with the FLOPs analysis. This suggests that our method has the potential to achieve comparable time complexity as BN through a well-engineered cuDNN implementation.

Table 6: Comparison of FLOPs per step of different normalization layers and total running times (in seconds) under different implementations on Split CIFAR-100 using ER-ACE with $|B| = 10$ and $|M| = 2000$ as the baseline approach. - to be exploited.

|  | BN | GN | CN | Ours |
|---|---|---|---|---|
| FLOPs/step | 49.20M | 49.18M | 86.09M | 49.32M |
|  | $1\times$ | $0.99\times$ | $1.75\times$ | $1.01\times$ |
| Time (Plain) | 430 | 440 | 535 | 489 |
|  | $1\times$ | $1.02\times$ | $1.24\times$ | $1.14\times$ |
| Time (cuDNN) | 324 | 322 | 376 | - |
|  | $1\times$ | $0.99\times$ | $1.16\times$ | - |

## D.2 Impacts of $r$

To show how the statistical weights of tasks affect performance, we vary the proportion of the current task in the batch (i.e., $r = N_t/N$) to different constants (with fixed $N_t = 10$, $N = 10, 20, 50, \dots$) and conduct experiments on Split CIFAR-10. It can be seen from Table 7 that the performance first increases rapidly, peaks at around $r = 1/10$ (i.e., the inverse of the total number of tasks), and then stabilizes. In particular, the accuracy with unfixed $r$ is comparable with the peak, which supports our Corollary 3. Since the total number of tasks is generally unavailable in continual learning, using an unfixed $r$ w.r.t $1/T$ is more practical.

Table 7: Performance comparison of different $r$ setups on Split CIFAR-10. $T$ refers to the number of seen tasks that increases over time.

| $r$ | 1 | 1/2 | 1/5 | 1/10 | 1/12 | 1/15 | 1/20 | 1/T |
|---|---|---|---|---|---|---|---|---|
| Task-IL ACC | 23.58 | 55.65 | 61.81 | **62.91** | 61.49 | 60.36 | 58.83 | 61.61 |
| Class-IL ACC | 3.05 | 17.33 | 25.26 | **25.50** | 25.07 | 24.04 | 23.83 | 24.26 |

## D.3 Impacts of Memory Buffer Selection Strategies

Fig. 5 shows that our method obtains substantial improvements over both reservoir sampling and ring buffer, which demonstrates the robustness of AdaB$^2$N to the memory buffer selection strategy.

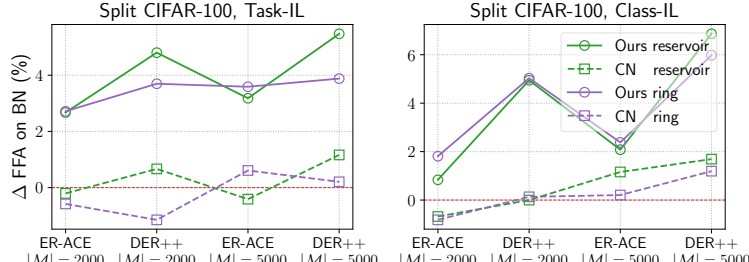

Figure 5: Performance improvements (i.e., $\Delta$ FAA) for online continual learning over different selection strategies of memory buffer and different memory sizes $|\mathcal{M}|$ (with batch size $|B| = 10$).

## D.4 Fine-tuning and Joint Training Performance

To establish a simple baseline and upper bound, we present the fine-tuning (FT) and joint training (JT) performance results of different normalization approaches in Table 8. Notably, Ours-FT consistently exhibits better performance than both BN-FT and CN-FT in the absence of replay data. On the other hand, Ours-JT does not outperform BN-JT, indicating that our performance improvements are indeed specific to continual learning.

Table 8: Fine-tuning and joint training performance on Split CIFAR-10 with $|B| = 10$.

| | Fine-tuning | | Joint tuning | |
| --- | --- | --- | --- | --- |
| | Task-IL ACC | Class-IL ACC | Task-IL ACC | Class-IL ACC |
| BN | 34.97±2.42 | 5.82±0.37 | **76.80±1.48** | 41.36±2.16 |
| CN | 34.48±0.88 | 4.68±0.59 | 76.41±0.61 | **41.46±0.61** |
| Ours | **36.31±1.72** | **5.95±0.88** | 75.79±1.31 | 40.14±1.74 |

## D.5 Scalability Analysis

In order to evaluate the scalability of our proposed method, we perform an additional experiment with ImageNet-sub [10], which includes 100 classes randomly selected from ImageNet of size $224 \times 224$. We split them into 10 incremental phases for online continual learning (ER, $|B| = 32$, $|M| = 2000$), and reuse the hyperparameters ($\lambda$ and $\kappa$) of Split Mini-ImageNet. As shown in Table 9, the Task-IL/Class-IL accuracy of our approach is clearly better than BN, demonstrating scalability on larger datasets. Furthermore, we perform experiments focused on online domain-incremental learning (Domain-IL) to evaluate its performance in handling more subtle distribution shifts. Specifically, we employ two datasets: Permuted MNIST [13], which applies random permutations to MNIST digit pixels to create 20 subsequent tasks, and DomainNet [18], comprising 345-class images from six distinct domains. Table 9 shows that our approach can clearly improve batch normalization on both Permuted MNIST and DomainNet, thanks to a better resolution of balance and adaptation.

Table 9: Performance on scaling to larger datasets and handling more subtle distribution shifts.

| | ImageNet-sub | | Permuted MNIST | DomainNet |
| --- | --- | --- | --- | --- |
| | Task-IL | Class-IL | Domain-IL | |
| BN | 72.92 | 16.86 | 64.89 | 7.38 |
| CN | 76.48 | 21.88 | 65.97 | **10.80** |
| Ours | **77.24** | **22.08** | **77.15** | *7.81* |

## D.6 Statistical Significance

We perform non-parametric paired t-test (i.e., the Wilcoxon signed-rank test) to evaluate whether AdaB$^2$N exhibits statistically significant superiority over comparable baselines in both online and offline scenarios, as well as a combination of both. The null hypothesis $H_0$ states that the mean performance difference between AdaB$^2$N and the alternate baseline is zero for a given scenario, while the alternative hypothesis $H_1$ states a non-zero difference. It is worth noting that we conduct two-tailed tests to avoid making an assumption that AdaB$^2$N consistently outperforms the baseline. Our samples are collected from all the results presented in the manuscript. The resulting $p$-values for these tests, as detailed in Table 10, demonstrate that AdaB$^2$N outperforms BN and CN statistically significantly across online, offline, and combined online and offline scenarios.

Table 10: Non-parametric paired t-test (the Wilcoxon signed-rank test) on whether AdaB$^2$N is statistically significantly better than the baselines ($p < 0.05$).

| | Online | | Offline | | Both | |
| --- | --- | --- | --- | --- | --- | --- |
| | BN | CN | BN | CN | BN | CN |
| $p$-value | 8.0e-13 | 5.2e-11 | 1.6e-02 | 1.5e-05 | 1.0e-14 | 1.0e-13 |
| Is AdaB$^2$N statistically significantly better? | ✓ | ✓ | ✓ | ✓ | ✓ | ✓ |

## D.7 Forgetting Measure

Table 11 shows that our method is generally the lowest in terms of forgetting measure.

Table 11: Forgetting measure (↓) of **online task-incremental learning** with batch size $|B| = 10$. We use **bold**, underline, and *italic* to indicate the first, second, and third best results respectively.

| Method | Split CIFAR-10 | | Split CIFAR-100 | | Split Mini-ImageNet | |
|---|---|---|---|---|---|---|
| | $\mathcal{M}$=500 | $\mathcal{M}$=2000 | $\mathcal{M}$=2000 | $\mathcal{M}$=5000 | $\mathcal{M}$=2000 | $\mathcal{M}$=5000 |
| ER-ACE w/ BN | 3.31±0.95 | 1.76±1.84 | 2.15±1.11 | 1.71±0.98 | *3.14±1.42* | *2.80±1.07* |
| ER-ACE w/ LN | 4.03±3.25 | *0.99±1.43* | 2.92±0.81 | 1.46±0.44 | 2.93±0.08 | 2.53±0.93 |
| ER-ACE w/ IN | 1.74±1.04 | 1.76±0.96 | *2.11±0.61* | **1.41±0.63** | 3.86±0.59 | 5.07±1.20 |
| ER-ACE w/ GN | 1.30±0.36 | 1.73±1.61 | **0.86±0.26** | 3.21±0.96 | 4.13±1.98 | 3.07±1.44 |
| ER-ACE w/ CN | *1.67±0.25* | 0.90±0.51 | 2.53±0.53 | 2.03±0.33 | 3.72±1.26 | 4.21±1.37 |
| ER-ACE w/ Ours | **1.22±0.60** | **0.84±0.78** | 1.81±0.66 | *1.58±1.35* | **2.21±2.38** | **1.94±0.87** |
| DER++ w/ BN | 2.35±0.73 | **0.22±0.03** | 1.26±0.42 | *1.31±1.07* | **1.97±0.94** | 2.60±0.32 |
| DER++ w/ LN | 1.61±1.05 | *0.65±0.22* | 2.02±1.28 | 1.88±1.41 | 3.16±0.70 | *2.95±0.61* |
| DER++ w/ IN | 1.90±0.47 | 0.68±0.62 | 2.91±1.89 | 1.78±1.03 | *3.46±0.56* | 3.64±0.40 |
| DER++ w/ GN | **1.42±1.03** | 0.89±1.05 | 2.32±1.45 | 1.41±0.63 | 3.88±1.52 | 3.70±1.29 |
| DER++ w/ CN | 3.89±0.52 | 2.05±0.40 | *1.63±0.77* | 1.07±0.28 | 4.04±1.17 | 3.85±1.08 |
| DER++ w/ Ours | *1.81±1.74* | 0.38±0.17 | **0.92±0.58** | **0.73±0.41** | 3.65±1.49 | **1.36±0.32** |

## D.8  Full Results for Dynamics of Normalization Statistics

We provide the norm dynamics of the batch statistics and population statistics for all normalization layer in Figs. 6 and 7. For better illustration, each curve in the figures indicates the mean after Gaussian smoothing with a kernel size of 20 and the shaded area indicates the $0.05 \times$ variance. It can be seen that our method roughly tracks the joint training (JT) on many layers (e.g., 2, 3, 5, 7, 8, 12, 15, 17, 18, 19).

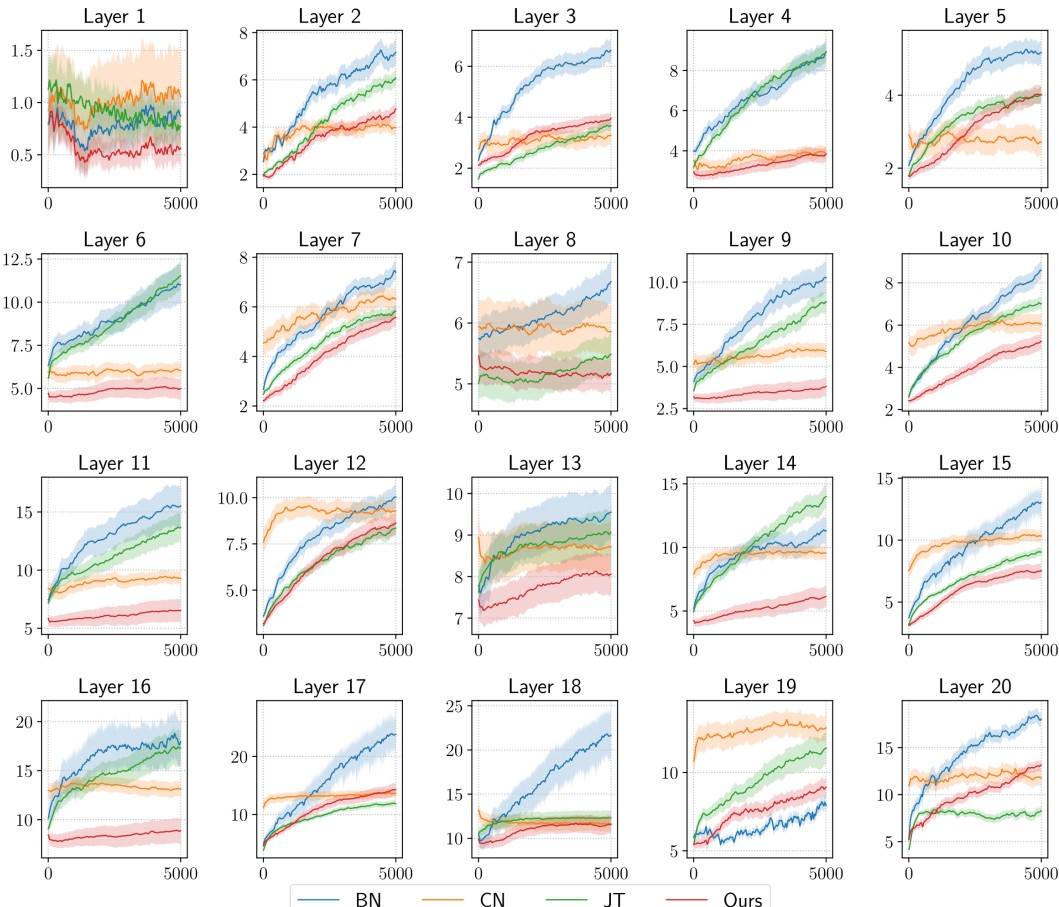

Figure 6: Norm dynamics of the batch statistics of all normalization layer.

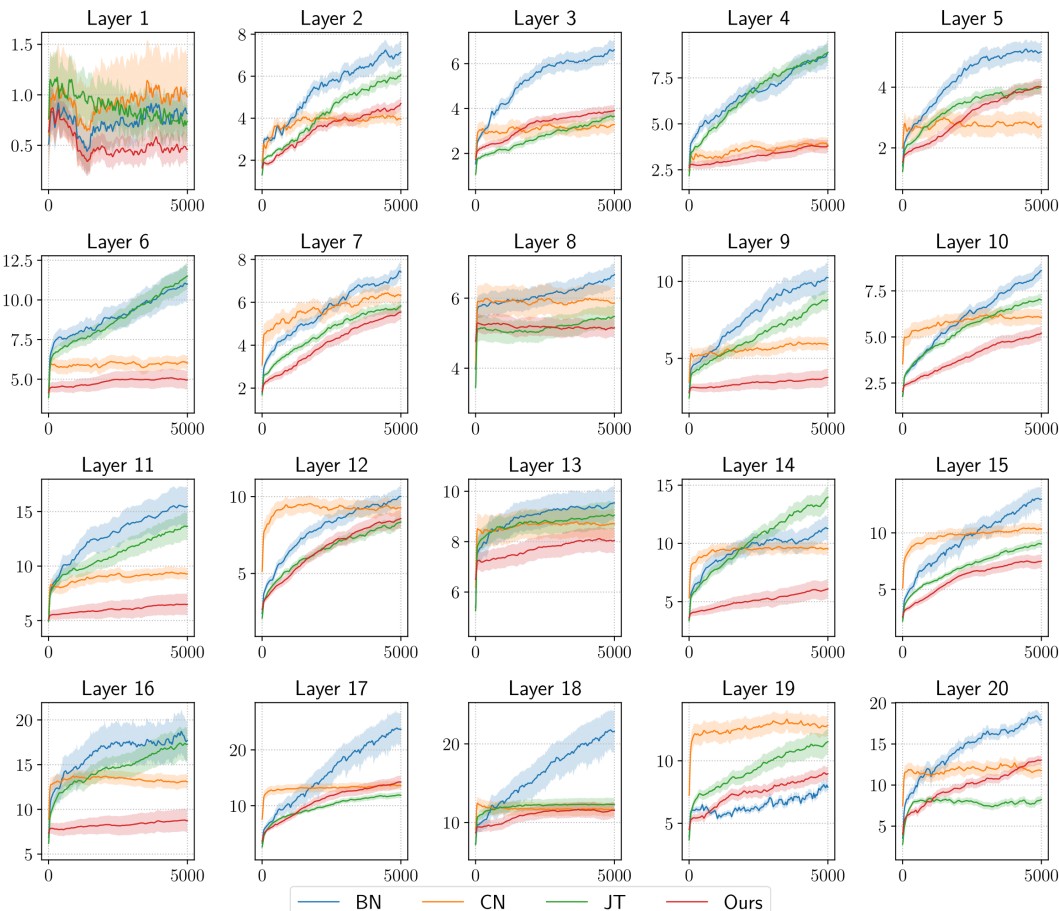

Figure 7: Norm dynamics of the population statistics of all normalization layer.

