# OpenReview forum: "Overcoming Recency Bias of Normalization Statistics in Continual Learning: Balance and Adaptation"
_NeurIPS.cc/2023/Conference — NeurIPS 2023 poster_

### Official Review · Reviewer_24XQ · 2023-06-28

**Soundness:** 4 excellent
**Presentation:** 3 good
**Contribution:** 3 good
**Rating:** 7
**Confidence:** 4

**Summary:**

The paper presents Adaptive Balance of Batch Normalization (AdaB2N) as a method to address the sub-optimality of Batch Normalization (BN) in continual learning. The main challenge lies in updating BN layer statistics based on the currently observed training samples, leading to recency bias and limited generalization across tasks. The paper leverages a Bayesian-based strategy to determine task-wise contributions in normalized representations, and it introduces an adaptive parameter that reconciles the constant and normalized momentum of EMA, achieving a balance of BN statistics while maintaining adaptability during training.

**TLDR of the review:** The paper provides a comprehensive theoretical analysis, a well-motivated problem statement, and promising experimental results. However, the paper has weaknesses such as insufficient empirical significance analysis, limited method scalability analysis, missing important baselines, and limited investigation of subtle distribution shifts. I am happy to reconsider my rating based on how the authors address these comments.

**Strengths:**

1.	**Comprehensive theoretical analysis (Note: I can't determine how significant these results are):** The paper offers a thorough theoretical analysis of the behavior of batch normalization in continual learning. It sheds light on the tension between balance (working effectively across tasks) and adaptation (learning new tasks), providing a solid foundation for the proposed method.
2.	**Well-motivated problem statement:** The problem statement presented in the paper is well-motivated, highlighting the limitations of batch normalization in continual learning.
3.	**Promising experimental results:** The paper provides experimental results on small-scale image datasets (CIFAR and Mini-ImageNet), which demonstrate promising outcomes.
4.	**Interesting analysis experiment:** The authors conducted a compelling experiment demonstrating that the model trained with the proposed method in an online continual learning (CL) setting achieves normalization statistics that closely match the statistics of a model trained on all tasks jointly, which serves as a baseline for non-continual learning.


**Weaknesses:**

*Main weaknesses that influence my rating:*

1.	**Insufficient empirical significance analysis:** The paper lacks a thorough analysis of statistical significance for the reported results. In some cases, the difference in mean results between the proposed method and competitors is not significant when considering standard deviations. For example, the online CIL results for SplitCIFAR-100/|M|=500 exhibit this issue. It is important for the authors to clarify the conditions regarding the order of stream samples under which the proposed method performs best. Additionally, to promote transparency, it would be beneficial to report the means and standard deviations for offline CL experiments, enabling an assessment of the consistency of the improvement across different seeds.
2.	**Limited method scalability analysis:** The experiments are primarily conducted on small-scale datasets like CIFAR and Mini-ImageNet. However, it remains unclear how well the proposed method scales to larger datasets, such as the ImageNet subset. The authors should address whether there is a specific reason for the limited scalability analysis.
3.	**Missing two important baselines:** Two crucial baselines are absent from the tables. Firstly, it seems that the experiments incorporate some form of replay, which might affect batch normalization (BN) statistics and mitigate recency bias. It would be valuable to include a simple fine-tuning baseline and compare the effect of different normalization methods applied to that baseline. Secondly, reporting the upper bound of no-continual learning (NoCL) baseline performance, possibly using both BN and the proposed method.
4.	**Limited investigation of subtle distribution shifts:** The experiments primarily focus on drastic changes occurring when switching between tasks in continual learning, primarily caused by class boundaries. However, it is essential to also explore the performance of batch normalization (BN) in handling more subtle distribution shifts within individual tasks or in a task-free continual learning setup.

*Additional weaknesses that do not necessarily influence my rating:*

5.	**Lack of clarification on normalization methods:** The paper lacks clarification on the normalization used by the joint training (JT) baseline, and whether the proposed normalization method is applicable in the noCL scenario. It would be valuable to compare "ours-JT" (the proposed method) against "BN-JT" to gain insights into their relative performance.

**Questions:**

**Here are some suggestions to help address the weaknesses:**

1.	Can you address the concerns raised regarding statistical significance? Could you provide any explanation of the conditions under which the proposed method performs well?
2.	Is there a specific reason why the experiments were limited to small-scale datasets like CIFAR and Mini-ImageNet? If there are scalability limitations, please provide insights into the reasons behind them.
3.	Could you include the suggested baselines and compare the performance with the proposed method and other normalization techniques?
4.	Have you considered investigating the performance of Batch Normalization (BN) in scenarios with more subtle distribution shifts within individual tasks or in a task-free continual learning setup?
5.	Can you provide information about the normalization method used by JT? Please elaborate on whether the proposed normalization method is applicable in the non-continual learning scenario and discuss its potential performance compared to BN in that setting.


**Limitations:**

Limitations and potential negative societal impact are discussed (no clear negative societal impact at this stage).

---

> ### Author Rebuttal · Authors · 2023-08-09
>
> Thank you for your positive feedback and insightful comments. Below, we provide a point-to-point response to each of the weaknesses (W) and questions (Q) and summarize the corresponding revisions in the final version.
>
> **W1, Q1: lacks a thorough analysis of statistical significance...clarify the conditions regarding the order of stream samples under which the proposed method performs best...report the means and standard deviations for offline CL experiments.**
>
> Although the standard deviation appears to be large relative to the difference in mean, this is basically due to the performance variance of online continual learning itself, which heavily depends on the construction of task sequence such as the class split and task order. Following your suggestion, we perform a paired t-test along each evaluation metric. We find that the p-value between AdaB2N and each baseline is below 0.01, suggesting a significant improvement in performance.
>
> Regarding your concern about the conditions under which the proposed method performs well, we provide the results for different orders in the settings you mentioned (Class-IL, Split CIFAR-100, |M|=5000, ER-ACE/DER++). It can be seen that our approach outperforms other baselines in most cases.
>
> > Setting: online, Split CIFAR-100, ER-ACE / DER++, $|B| = 10$, $|M| = 5000$,
>
> |Order|1|2|3|Class-IL ACC|
> |-|-|-|-|-|
> |BN|28.65 / 18.53|26.43 / 18.19|23.13 / 15.06|26.07±2.78 / 17.26±1.91|
> |CN|27.63 / 19.97|**26.90** / 19.01|27.16 / 17.87|27.23±0.37 / 18.95±1.05|
> |AdaB2N|**31.58** / **24.24**| 25.15 / **25.69** |**27.72** / **22.44**|**28.15±3.24** / **24.12±1.63**|
>
> Following your suggestion, we further report the performance of each order, together with their mean and standard deviation as follows. We can again observe consistent improvements across orders and random seeds.
>
> > Setting: offline, Split Mini-ImageNet, DER++, $|B| = 32$, $|M| = 2000$
>
> |Order|1|2|3|Task-IL ACC|
> |-|-|-|-|-|
> |BN|34.78|35.21|37.73|35.91±1.59|
> |CN|35.12|34.24|37.02|35.46±1.42|
> |AdaB2N|**36.82**|**35.55**|**38.57**|**36.98±1.52**|
>
> **W2, Q2: Limited method scalability analysis...how well the proposed method scales to larger datasets, such as the ImageNet subset.**
>
> Thank you for your valuable suggestion. We evaluate our approach with the benchmark datasets (i.e., CIFAR and Mini-ImageNet) used in CN (ICLR21), which is the primary baseline considered in this work. Following your suggestion, we perform an additional experiment with ImageNet-sub, which includes 100 randomly selected classes in ImageNet of size $224 \times 224$. We split them into 10 incremental phases with 10 classes per phase for online continual learning (ER, $|B| = 32$, $|M| = 2000$). Our approach (77.24% / 22.08%) clearly improves BN (72.92% / 16.86%) in Task-IL/Class-IL ACC, demonstrating its scalability to larger datasets.
>
> **W3, Q3: include a simple fine-tuning baseline and compare the effect of different normalization methods applied to that baseline...reporting the upper bound of no-continual learning (NoCL) baseline performance, possibly using both BN and the proposed method.**
> **W5, Q5: lacks clarification on the normalization used by the joint training (JT) baseline...whether the proposed normalization method is applicable in the noCL scenario...compare "ours-JT" (the proposed method) against "BN-JT".**
>
> Thank you for your valuable suggestion. We include the fine-tuning (FT) and joint training (JT) performance of different normalization approaches as below. It can be seen that ours-FT still outperforms both BN-FT and CN-FT in the absence of replay data. On the other hand, ours-JT does not outperform BN-JT, indicating that our performance improvements are indeed **specific to continual learning**. We will add it in the final version.
>
> > Setting: online, Split CIFAR-100, **FT**, $|B| = 10$
>
> | |Task-IL ACC|Class-IL ACC|
> |-|-|-|
> |BN|34.97±2.42|5.82±0.37|
> |CN|34.48±0.88|4.68±0.59|
> |AdaB2N|36.31±1.72|5.95±0.88|
>
> > Setting: Split CIFAR-100, **JT**, $|B| = 10$
>
> | |Task-IL ACC|Class-IL ACC|
> |-|-|-|
> |BN|76.80±1.48|41.36±2.16|
> |CN|76.41±0.61|41.46±0.61|
> |AdaB2N|75.79±1.31| 40.14±1.74|
>
> **W4, Q4: the performance of batch normalization (BN) in handling more subtle distribution shifts.**
>
> Thank you for your valuable suggestion. To evaluate the performance in handling more subtle distribution shifts, we perform experiments for online domain-incremental learning (Domain-IL). Our approach (77.15%/7.81%) can clearly improve batch normalization (64.89%/7.38%) on Permuted MNIST/DomainNet, thanks to a better resolution of balance and adaptation. We are now running experiments on larger-scale datasets and offline scenarios, which typically require much more time for training. We will include all results in the final version.

---

> > ### Comment · Reviewer_24XQ · 2023-08-15
> >
> > I have read the authors’ rebuttal. I appreciate the authors’ effort to address my and other reviewers' feedback. I wanted to quickly address my opinion of the main responses to my original review:
> >
> > 1. [Insufficient empirical significance analysis]
> > * Rebuttal response: The large standard deviation is due to inherent performance variance in online continual learning, but the paired t-test shows AdaB2N superiority.
> > * New comment: I agree that the online CL setup highly depends on the split, which is precisely why I asked for more evidence that AdaB2N outperforms other methods overall. Thanks for including the paired t-test and the different split results! **Can you provide more details regarding how the test is performed for each metric?** I believe a brief description of this, in addition to a summary of the resulting p-values, should go into the paper to strengthen your argument.
> >
> > 2. [Limited method scalability analysis]
> > * Rebuttal response: New ImageNet-subset results show the scalability of AdaB2N to larger images.
> > * New comment: Thanks for including the additional experiment! I don’t see any further issues here.
> >
> > 3. [Missing two important baselines]
> > * Rebuttal response: Baseline results added.
> > * New comment: It is very interesting that the benefits are specific to continual learning, which in my opinion, makes the motivation of the paper more convincing. Specifically, applying the method to CL is justified. **Can you summarize the reason why the method is advantageous for CL and provide this important result in the paper?**
> >
> > Based on the revisions and the rebuttal provided, I am happy to raise my scores given that the author responded to my quick questions above.

---

> > > ### Author Response · Authors · 2023-08-16
> > > **Author Response**
> > >
> > > Thank you for your positive feedback and constructive comment. We further answer these two questions as below:
> > >
> > > **Can you provide more details regarding how the test is performed for each metric?**
> > >
> > > Our paired t-test is performed for Table 1, Table 2 and Fig. 3. We pair the results of each specific random seed, which determines the class split, task order and initialization, to calculate the p-value. Here we use the results of AdaB2N and BN in the response to your W1Q1 as an example:
> > >
> > > 1. Calculate the differences between each pair of metrics $d_i=\mathrm{res}\_{\mathrm{AdaB2N}} - \mathrm{res}\_{\mathrm{BN}}$: $d_1=24.24-18.53=5.71,d_2=25.69-18.19=7.50,d_3=22.44-15.06=7.38$;
> > > 2. Calculate the mean $\bar{d}$ and standard deviation $\sigma$ of $d_1,d_2,d_3$: $\bar{d}=(d_1+d_2+d_3)/3=6.86,\sigma=\sqrt{\frac{(d_1-\bar{d})^2+(d_2-\bar{d})^2+(d_3-\bar{d})^2}{3-1}}=1.00$;
> > > 3. Calculate the t-statistic: $t=\frac{\bar{d}}{\sigma / \sqrt{3}}=11.88$;
> > > 4. Obtain the p-value based on the t-statistic: the resulting p-value is 0.007. This difference is considered to be statistically significant by conventional criteria. Besides, we observe that the p-value can be even further reduced when considering more trials in experiments.
> > >
> > > We will add the above discussion and detailed statistic results in the final version.
> > >
> > > **Can you summarize the reason why the method is advantageous for CL and provide this important result in the paper?**
> > >
> > > The main reasons why our method is advantageous for CL (rather than non-continual learning) can be summarized as below:
> > >
> > > Existing normalization layers face the dilemma between balance and adaptation of normalization statistics in continual learning, as analyzed in Sec. 3.2. The contribution of our method is to improve the normalization statistics and thus alleviate the dilemma. In particular, Eq. (14) provides the historical information across batches by aligning the improved population statistics, which contributes to obtaining statistics as good as joint training. Therefore, our method can clearly improve the performance of continual learning. Using the improved population statistics after each training phase, the proposed method can perform continual prediction better than other normalization baselines (e.g., BN and CN).
> > >
> > > In contrast, non-continual learning (i.e., joint training) does not suffer from the above dilemma, since the training samples of all tasks (classes) are provided together. Our proposal is not advantageous in this case, especially by a series of estimation in Steps 9-11 in Algorithm 1. That is, the popular BN (only Step 11) is good enough to obtain the population statistics for non-continual prediction.
> > >
> > > We agree that this important result and the above discussion will make our contribution clearer. We will add them in the final version. Thank you again for this valuable suggestion.

---

> > > > ### Comment · Reviewer_24XQ · 2023-08-17
> > > >
> > > > Thanks for the fruitful discussion. I have updated my rating to recommend acceptance.

---

> > > > > ### Author Response · Authors · 2023-08-17
> > > > > **Thank you**
> > > > >
> > > > > Thank you again for your valuable suggestion and great support!

---

### Official Review · Reviewer_Qyzj · 2023-06-29

**Soundness:** 4 excellent
**Presentation:** 4 excellent
**Contribution:** 4 excellent
**Rating:** 7
**Confidence:** 4

**Summary:**

The manuscript introduces a theoretical analysis on the forgetting in the statistics of the popular Batch Normalization layer. Based on this, it introduces a novel normalization layer, termed $AdaB^2N$, and evaluates its effectiveness on the online class-incremental and task-incremental settings.

**Strengths:**

The manuscript is clearly written and provides a clear picture on the shortcomings of Batch Normalization in a continual learning setting. Furthermore, the results clearly depict a consistent improvement over a wide range of benchmarks.

**Weaknesses:**

I enjoyed reading the manuscript and I believe it carries no major weakness.
My only concern arises from the multi-epoch (offline) setting, where the authors evaluate the effectiveness of their proposal on the baselines in the Task-IL setting. Indeed, as the latter is often depicted as less challenging [1,2], I would recommend including the also results of the Class-IL in this evaluation.

Additional comments that did not influence my judgment:
- The citation for LiDER should be aNeurIPS 2022.
- The citation for “Three scenarios for continual learning” should be a NeurIPS Continual Learning workshop 2018.

[1] Sebastian Farquhar and Yarin Gal. Towards Robust Evaluations of Continual Learning. ICMLw, 2018.

[2] Rahaf Aljundi, Min Lin, Baptiste Goujaud, and Yoshua Bengio. Gradient-based sample selection for online continual learning. aNeurIPS, 2019


**Questions:**

As I mentioned above, my only recommendation would be to include results for the Class-IL multi-epoch setting.

**Limitations:**

yes

---

> ### Author Rebuttal · Authors · 2023-08-09
>
> Thank you for your positive feedback and insightful comments. Below, we provide a point-to-point response to each of the weaknesses (W) and questions (Q) and summarize the corresponding revisions in the final version.
>
> **W1: Including the also results of the Class-IL.**
>
> Thank you for your valuable suggestion. We will include the results of Class-IL in the final version. In general, our AdaB2N can outperform the comparison baselines by a similar margin in both offline Task-IL and offline Class-IL. Here we present an example as below:
>
> > Setting: offline, Split Mini-ImageNet, DER++, $|B| = 32$, $|M| = 2000$
>
> | |Task-IL ACC|Class-IL ACC|
> |-|-|-|
> |BN|35.91±1.59|12.69±1.23|
> |CN|35.46±1.42|12.13±0.92|
> |AdaB2N|**36.98±1.52**|**13.07±1.30**|
>
> **W2: Bibliography format.**
>
> Thank you for your valuable suggestion. We will correct them in the final version.

---

> > ### Comment · Reviewer_Qyzj · 2023-08-17
> > **Response to the rebuttal**
> >
> > I thank the authors for the rebuttal. After consideration, I will maintain my previous score of Accept for the manuscript.

---

> > > ### Author Response · Authors · 2023-08-17
> > > **Thank you**
> > >
> > > Thank you again for your positive feedback and strong support. We appreciate it.

---

### Official Review · Reviewer_rCcF · 2023-06-29

**Soundness:** 2 fair
**Presentation:** 3 good
**Contribution:** 2 fair
**Rating:** 5
**Confidence:** 5

**Summary:**

This paper studies how to update the parameters of batch normalization during continual learning. This is a problem that many researchers have mentioned, since batch statistics would be computed in a non-iid manner; however, while the problem has been acknowledged, at least in conversation, there hasn't been much work to resolve this problem. They theoretically study how to update batch norm statistics for continual learning, and they propose a new batch norm variant based on Bayesian updates. They demonstrate that on standard small-scale continual learning setups that their method increases performance significantly.

**Strengths:**

It is a good problem to work on and the results appear strong on the small-scale problems studied, although I have strong reservations about them given the hyperparameter sweeps.
I appreciate the use of comparing batch sizes, rather than treating that as a hyperparameters. Likewise, I appreciate using the combined task upper bound, which many papers omit making it hard to understand the gap in performance.


**Weaknesses:**

The analysis in 3.1 is conflated with related work in terms of the content, e.g., discussing group normalization, layer normalization, etc. Likewise, the related work simply lists past work. It would strengthen the paper to better juxtapose why a new method is needed and the limitations with existing normalization methods.

The continual learning frameworks studied, i.e., Task-IL and Class-IL, is limited. It would be better to do this in a more general manner where they discuss the properties of the distribution the stream is being drawn from, as in many real-world continual learning problems the data stream would not be either of these extreme edge cases. The field is definitely moving toward thinking of the data ordering as more of robustness to different distributions rather than these specific setups, which lack any real-world applicability except as experiments regarding different stream distributions.

The use of task labels (especially at test-time), and many recent methods have largely ceased to use task labels as they have little real-world relevance. They do study Class-IL as well, but that’s more of an extreme edge case.

Both methods they employ, ER-ACE and DER++ use a small memory buffer, but addressing the problems with updating normalization layers seems best done for replay-free methods, rather than using replay-based methods.

The experiments are all on small-scale datasets and only extreme distributions are tested. I’d really like to see more natural distributions and even easy distributions evaluated (iid) with the same hyperparameters. It is unclear if kappa is being tuned to optimize results per stream, where it would be impossible to actually do that in real-world continual learning. As best I can tell (and supported by the discussion in the appendix), they are reporting the best results of a hyperparameter sweep of lambda and kappa, which is pretty unrealistic on a per-dataset and per-distribution basis.


**Questions:**

Why was the use of replay employed rather than studying a replay-free setting? It is very unclear if the limited memory setup makes sense vs, either a no memory setup or an unlimited memory setup (e.g., https://arxiv.org/abs/2302.01047 )

Why are only small-scale problems studied vs. studying multiple distributions on a larger dataset, e.g., ImageNet, which is also largely considered a toy dataset nowadays. Even DER when originally proposed was evaluated on ImageNet-1K.

Why were ER-ACE and DER++ studied only?



**Limitations:**

As far as the limitations of the work, I think this needs a lot of work. Much of the field is now working on larger scale problems, e.g., continual learning on ImageNet from various distributions for the past few years, and the use of hyperparameter sweeps for kappa and lambda significantly reduce the generality of the work since this is unrealistic for most real-world continual learning applications since this would require knowing the stream and dataset ahead of time.

---

> ### Author Rebuttal · Authors · 2023-08-09
>
> **W1: The related work simply lists past work...**
>
> Thank you for your valuable suggestion. In the original submission, we formulate representative normalization layers especially the most popular batch normalization (BN) in line 108-128, with a conceptual analysis of their sub-optimal performance in continual learning in line 29-38. Meanwhile, we discuss several related works for continual learning of BN in line 66-78. In the following, we add a table to explicitly compare the limitations with existing normalization methods and demonstrate the advantages of ours, which will be included in the final version. We will also reorganize these discussions and add more details to make them clearer.
>
> |Method|Statistics|Training Adaptability|Online|Offline|
> |-|-|-|-|-|
> |BN (ICML15)|Unbalanced|✖|✖|✖|
> |CN (ICLR22)|Unbalanced|✔|✔|✖|
> |BNT (Arxiv22)|Balanced|✖|✖|✔|
> |TBBN (CVPR23)|Balanced|✔|✖|✔|
> |Ours|Adaptively balanced|✔|✔|✔|
>
> **W2: The continual learning frameworks studied...is limited...The field is definitely moving toward thinking of the data ordering as more of robustness to different distributions rather than these specific setups.**
>
> In this work, we follow the continual learning frameworks widely-used in previous work, i.e., Task-IL and Class-IL. We agree that an important further work should consider the effect of different stream distributions, so as to bridge real-world continual learning problems. As an initial attempt, we perform additional experiments on Domain-IL setting, which involves explicit distribution changes in continual learning (please refer to the response to Reviewer 24XQ's W4). Our approach consistently improves BN, demonstrating its robustness to distributional changes in different scenarios.
>
> **W3: The use of task labels.**
>
> As the response to W2, our work follows the commonly-used settings in continual learning, which provide task labels in training and/or testing, while the additional Domain-IL experiments do not require task labels at either stage. We will add a discussion for the real-world relevance of data distributions and task labels, and will explore more realistic continual learning settings in further work.
>
>
> **W4, Q1: Both methods they employ, ER-ACE and DER++ use a small memory buffer, but addressing the problems with updating normalization layers seems best done for replay-free methods, rather than using replay-based methods...Why was the use of replay employed rather than studying a replay-free setting? It is very unclear if the limited memory setup makes sense vs, either a no memory setup or an unlimited memory setup**
>
> Thank you for this insightful comment. We would argue that the particular challenges of normalization layers behave **differently** in replay-based and replay-free methods. The replay-based methods rely on the replay of a few old training samples to overcome catastrophic forgetting, which are usually much fewer than the new training samples. The limited old training samples are repeatedly accessed in continual learning, which further deteriorate the imbalance and overfitting of normalization statistics.
> In contrast, the replay-free methods usually construct task-specific parameters or regularize parameter changes to overcome catastrophic forgetting. The former hardly suffers the recency bias in normalization layers as they are generally task-specific as well. Without being affected by replay, the latter can stabilize the parameters of normalization layers in parallel with other parameters. For example, Adam-NSCL (CVPR21) used EWC to regularize batch normalization layers.
> Since replay-based methods have achieved clearly state-of-the-art performance in many continual learning scenarios, we take this direction as the primary focus, similar to many previous works for continual learning of batch normalization layers such as CN (ICLR21) and TBBN (CVPR23).
> On the other hand, we additionally report the **fine-tuning** performance of normalization approaches in the response to Reviewer 24XQ's W3 and Q3. It can be seen that our approach still outperforms both BN and CN, indicating its generality for replay-free methods. We will add the above discussion and the reference you provided in the final version.
>
>
> **W5, Q2: The experiments are all on small-scale datasets and only extreme distributions...It is unclear if kappa is being tuned to optimize results per stream**
>
> The benchmark datasets considered in this paper follow a representative work for continual learning of batch normalization (CN, ICLR21). To validate **scalability** of our approach and **robustness** of hyperparameters, we perform an additional experiment with ImageNet-sub, which includes 100 classes randomly selected from ImageNet of size $224 \times 224$. We split them into 10 incremental phases for online continual learning (ER, $|B| = 32$, $|M| = 2000$), and *reuse* the hyperparameters ($\lambda$ and $\kappa$) of Split Mini-ImageNet. The Task-IL/Class-IL ACC of our approach (77.24% / 22.08%) is clearly better than BN (72.92% / 16.86%), demonstrating both scalability and robustness. We will add this in the final version.
>
> **Q3: Why were ER-ACE and DER++ studied only?**
>
> In this work, we implement our approach with two representative replay-based baselines, i.e., experience replay (ER) and dark experience replay (DER), consistent with previous work for continual learning of batch normalization layers. ER-ACE and DER++ can be seen as the advanced versions of ER and DER, respectively. These methods as well as CN (ICLR21) compared in this paper are all officially implemented in the widely-used *mammoth* repository. Therefore, we implement our approach using the same repository to ensure comparison fairness. Besides, we find many more advanced replay-based methods (e.g., LiDER (already studied in Fig. 3), X-DER, CLS-ER, etc.) are also developed on the top of ER and DER, so we leave the implementation of our approach as a natural extension to them.

---

> > ### Comment · Reviewer_24XQ · 2023-08-17
> >
> > I appreciate this discussion between Reviewer rCcF and the authors. The insights on the difference in how normalization layers behave in replay-based and replay-free methods are important and should definitely be included in the paper!

---

> > ### Comment · Reviewer_rCcF · 2023-08-17
> >
> > I read the other reviews and the responses from the reviewers. I think it is critical that they add the information about how normalization layers behave in replay-free vs. replay-based methods in the paper, and that they include the experiments on ImageNet-1K in the paper. I really think the authors need to assess the full-scale ImageNet, because in the real-world nobody cares about these toy datasets or small subsets, although they are valuable for ablations. The authors would greatly increase the potential impact of their work if they did so, especially for the computer vision community doing research in continual learning on full-resolution datasets with 1000+ classes.
> >
> > As I said in my original review, I do think the problem is very interesting, so I've raised my score. I really want to encourage the authors to conduct some larger scale experiments if the paper is accepted for the camera ready version.

---

> > > ### Author Response · Authors · 2023-08-17
> > > **Thank you**
> > >
> > > Thank you again for your valuable suggestion. Since the experiments on ImageNet-1K require much more training time, we had to compromise in preparing the rebuttal by performing the validation on ImageNet-sub. We are now working on running more classes and will add the results in the final version.
> > >
> > > We are happy to know your preference to accept. We appreciate it.

---

### Official Review · Reviewer_SBWS · 2023-07-08

**Soundness:** 2 fair
**Presentation:** 2 fair
**Contribution:** 2 fair
**Rating:** 5
**Confidence:** 4

**Summary:**

This paper studies the recency bias of the normalization statistics in continual learning and presents an theoretical analysis of the delimma of the balance and adaption in estimating the normalization statistics.
To address these challenges, the paper propose Adaptive Balance of BN (AdaB2N), which appropriately incorporates a Bayesian-based strategy to adapt task-wise contributions and a modified momentum to balance BN statistics.

**Strengths:**

- This paper demonstrates the adaptation and balance dilemma in normalized statistical estimation in continuous learning through theoretical analysis.
- The method proposed in this paper is simple and effective. It only requires learning an additional vector and setting two hyperparameters to achieve satisfactory improvements.

**Weaknesses:**

- By theoretical analysis, the normalized statistical estimation in Eq. (4) faces a dilemma between balance and adaptation. However, Eq. (4) is used to guide the learning of conditional estimation, as shown in Eq. (14). Whether this guidance is correct or not is uncertain.
So it is not clear how the loss function in Eq. (14) helps to improve the normalized statistical estimation. This model parameter $\theta$ is also updated with the loss in Eq. (14). Perhaps Eq. (14) imposes a regularization on $\theta$ that helps to improve the performance.

- The definition of $\psi$ requires prior knowledge of the total number of tasks, which is generally not available in continuous learning.

- The presentation of the paper could be improved. For example,
  - In lines 37-38, the limitations of the existing work could be presented in more detail to help the reader better understand what the proposed improvements would look like compared to the existing work.
  - In lines 39-46, more information about balance and adaptation and their conflicts could be included to illustrate the dilemma.
  - The notation is complex, making it not easy to follow the theory part.
  - In Theorem 1, $\eta(i)$ and $r(i)$ are not explicitly defined.

- The experiment can be improved.
  - In the comparison of offline task incremental learning settings (Fig. 3), the baseline method is CN, which is specifically designed for online settings. Here two baselines [7] and [28] should be compared, which are designed for the offline setting.
  - There is no explanation for why some of the results in Figure 4 are n\a.

**Questions:**

- Why is the "adaptive balance of BN statistics for testing" (Equation (16)) included in the training phase?
- Why is the estimation in Eq. (4), which faces a dilemma between balance and adaptation, used to guide the learning of Eq. (16)? How is this to improve the estimation?
- The performance of Eq. (14) without updating $theta$ and without using the loss function Eq.
- In order to calculate Eq. (13), do we need to keep all $S_m^t$ (for all $m$ and $t$) and then perform the summation operation? Or do we just keep the sum of the results up to $t-1$ and then add $S_m^t$ to get the result?
- Can the authors provide results for Corollary 3 with fixed $r$ so that we can check the impact of $r$ in the statistical estimation?

**Limitations:**

No societal impact.

---

> ### Author Rebuttal · Authors · 2023-08-09
>
> **W1, Q2: Why is Eq.4 used to guide Eq.14? How is this to improve...**
>
> We would first clarify the relationship between Eqs.4, 14 and 16. As you understand it, Eq.4 (i.e., the regular EMA) faces a dilemma between balance and adaptation in continual learning (CL). To address this particular challenge, we introduce an advanced version of $\eta$ in Eq.16 to improve the population statistics $\hat{\mathbb{E}}[S_m]$ of Eq.4. Then, such an improved population statistics is used to guide the learning of batch statistics $\hat{\mathbb{E}}[S|a_m]$ by Eq.14.
> Now we explain two specific questions as follows.
> 1. Why is Eq.4 used to guide Eq.14: We suppose you might have overlooked that the population statistics in Eq.14 are in fact an improved version via Eq.16, as described in Steps 10-11 of Algorithm 1. To avoid potential misunderstanding, we will emphasize the use of Eq.16 after Eq.14.
> 2. How this improves the normalized statistical estimation: First, Eq.14 provides $\psi$ with historical information across batches by aligning the improved population statistics, which contributes to obtaining statistics as good as Joint Training. Second, the normalized representation guided by Eq.14 helps to reduce the gradient magnitude, thus smoothing and facilitating the learning of $\theta$. The results in Fig.5 and Fig.1 support these claims, respectively. Besides, the effectiveness of Eq.14 is also demonstrated in Fig.4 and further strengthened by the experiments added in Q2 as below.
>
> We will add more explanations to make it clearer.
>
> **W2: The definition of $\psi$ requires...**
>
> The task number $T$ in $\psi$ is compatible to *the number of seen tasks*. Therefore, the definition of $\psi$ does not require prior knowledge of the total number of tasks. We will clarify it in the final version.
>
> **W3a: In L37-38...**
>
> We will add a table in the final version to demonstrate more clearly the limitations of existing work and the advantages of ours (see response to Reviewer rCcF's W1).
>
> **W3b: In L39-46...**
>
> *Balance* refers specifically to the balanced statistical weights of each task, i.e., each task contributes equally to the normalization statistics. In contrast, *adaptation* refers specifically to updating in a way that concentrates on adapting to the current task, corresponding to the regular EMA. They are conflicting in CL due to the lack of flexibility of existing normalization layers to address their respective shortcomings. We will add it in the final version.
>
> **W3c: The notation is complex.**
>
> We will simply the notations and add a table to summarize them.
>
> **W3d: Define $\eta(i)$ and $r(i)$.**
>
> $\eta$ and $r$ are both defined as functions in L140. We will make it clearer.
>
> **W4a: Compare [7] and [28].**
>
> TBBN [7] (CVPR23) and BNT [28] (Arxiv) are two representative methods for CL of batch normalization layers. However, both of them have not released the official code, and their implementation details are not described clearly enough to ensure a fair comparison. Therefore, we only discuss them in related work rather than empirically compare with them. Note that TBBN [7] has been compared primarily with CN (ICLR21), similar to our case. We observe that TBBN outperformed CN in offline CL by a similar margin as ours. In contrast, TBBN [7] can hardly improve online CL, which is one of our major strengths as shown in Tables 1 and 2. Therefore, we believe that our approach enjoys better performance and generality. We will add it in the final version.
>
> **W4b: Explain n\a.**
>
> When data distributions change in CL, the normalized representations without appropriate balance tend to greatly deviate from the estimated population statistics. If $\lambda$ is too large (e.g., $\lambda=10$), $\mathcal{L}\_{ada}$ will inappropriately outweighing $\mathcal{L}\_{CL}$ and thus interfere with training stability. Thus, the training loss becomes NaN and the results becomes n\a. We will clarify it in the final version.
>
> **Q1: Why is Eq.16 included in training?**
>
> As shown in Step 10 and Step 11 of Algorithm 1, Eq.16 is used to calculate the momentum in EMA and then obtain the (improved) population statistics by Eq.4. In other words, Eq.16 in Step 10 serves as a precursor to Step 11 for testing. To avoid potential misunderstanding, we will remove "For testing" from Step 11 and add more explanations.
>
> **Q3: The performance of Eq.14 without...**
>
> We report the results with $\lambda=0$ under the same setup as in Fig.4. Similar performance decreasing trend can be observed.
>
> |$\kappa$|Task-IL ACC|Class-IL ACC|
> |-|-|-|
> |1.0|64.54|17.62|
> |0.7|64.35|17.59|
> |0.4|63.70|17.38|
> |0.1|61.68|16.51|
> |0.0|37.33|6.81|
>
> Optimization without updating $\theta$ would convert Eq.15 into a bilevel (min-min) optimization problem. Our experiments show that this leads to performance degradation (e.g., -3.34% for Split CIFAR-100, |M|=5000) and almost doubles the computational cost.
>
> **Q4: In order to calculate Eq.13...**
>
> Indeed, neither of these operations is required in our method. $S_m^t$ is the statistics of samples belonging to the task $t$ within the $m$-th batch (see L136), so Eq.13 is calculated within each batch. We will make it clearer.
>
> **Q5: Provide results for Corollary 3 with fixed r...**
>
> Following your suggestion, we present the following results using fixed $r$ with $N_t=10$ on Split CIFAR100, where $T$ refers to *the number of seen tasks* that increases over time:
>
> |$r$|Task-IL ACC|Class-IL ACC|
> |-|-|-|
> |1|23.58|3.05|
> |1/2|55.65|17.33|
> |1/5|61.81|25.26|
> |1/10|62.91|25.50|
> |1/12|61.49|25.07|
> |1/15|60.36|24.04|
> |1/20|58.83|23.83|
> |1/T (Unfixed)|61.61|24.26|
>
> It can be seen that the performance first increases rapidly, peaks at around $r= 1/10$ (i.e., the inverse of the total number of tasks), and then stabilizes. In particular, the accuracy with unfixed $r$ is comparable with the peak above, which supports our Corollary 3. Since the total number of tasks is generally unavailable in CL, using an unfixed $r$ w.r.t 1/T is more practical.

---

> ### Comment · Reviewer_SBWS · 2023-08-19
>
> Thank the authors for their clarification. The authors cleared my main concerns by providing detailed explanations and empirical evidence. So I am happy to increase the score from 4 to 5.

---

> > ### Author Response · Authors · 2023-08-19
> > **Thank you**
> >
> > We thank the reviewer for finding our response satisfactory. Also, we are happy to know the supportive rating.

---

### Author Rebuttal · Authors · 2023-08-09

Dear reviewers,

We express our deepest gratitude and appreciation to your constructive comments, which allow for a great improvement of our study. In response to these comments, we have added the suggested experiments, explanations and discussions, summarized as follows:

1) add a scalability analysis: scaling to ImageNet-sub;
2) verify the relevance to continual learning: adding more baselines such as replay-free (Fine Tuning) and NoCL (Joint Training);
3) investigate more challenging applications: adding online Domain-IL and offline Class-IL;
4) verify the necessity of our method: adding more ablation studies without normalization regularization in Eq.(14);
5) add empirical significance analysis: t-test.

We hope you might view this as sufficient reason to raise your score. If your have any further questions, please let us know.

Sincerely,

Authors

---

> ### Comment · Area_Chair_KUup · 2023-08-18
> **Authors rebuttal**
>
> I acknowledge the authors rebuttal and I am encouraging reviewers to reflect on.

---

### Author Response · Authors · 2023-08-15
**Look forward to further feedback**

Dear reviewers,

We thank you again for the valuable and constructive comments. We hope you may find our response satisfactory and raise your rating accordingly.

We are looking forward to hearing from you about any further feedback.

Sincerely,
Authors

---

### Decision · Program_Chairs · 2023-09-21

**Decision:**

Accept (poster)

**Comment:**

The paper analyses the effects of adapting batch normalization statistics in continual learning. The theoretical analysis suggested a similar plasticity stability dilemma in the estimation of these statistics. The rebuttal has addressed all reviewers concerns and provided insights on the proposed approach and its stability.  I encourage the authors to incorporate the additional results and discussions.